# Automated design of protein-binding riboswitches for sensing human biomarkers in a cell-free expression system

Grace E. Vezeau[1], Lipika R. Gadila[2] & Howard M. Salis [1,2,3,4] ✉

Cell-free genetically encoded biosensors have been developed to detect small molecules and nucleic acids, but they have yet to be reliably engineered to detect proteins. Here we develop an automated platform to convert protein-binding RNA aptamers into riboswitch sensors that operate within low-cost cell-free assays. We demonstrate the platform by engineering 35 protein-sensing riboswitches for human monomeric C-reactive protein, human interleukin-32γ, and phage MS2 coat protein. The riboswitch sensors regulate output expression levels by up to 16-fold with input protein concentrations within the human serum range. We identify two distinct mechanisms governing riboswitch-mediated regulation of translation rates and leverage computational analysis to refine the protein-binding aptamer regions, improving design accuracy. Overall, we expand the cell-free sensor toolbox and demonstrate how computational design is used to develop protein-sensing riboswitches with future applications as low-cost medical diagnostics.

Synthetic biologists have created a wide variety of sensor systems to detect small molecules and nucleic acids[1–13]. Several of these sensors have been developed for usage in cell-free expression systems, where there is no barrier between the expression machinery and exogenously added bulky macromolecules that would otherwise be unable to pass through a cellular membrane[14–16]. Cell-free sensors are particularly useful as low-cost, portable diagnostic and field assays as they genetically encode their own detection machinery and do not require a cold chain during storage and distribution[14,15,17–22]. However, even though protein detection is a cornerstone of both modern medical diagnostics and biological research[23], there are only a few cell-free sensors that utilize gene regulation to detect proteins of interest[24–26].

Currently, measuring protein titers is widely carried out using immunoassays (e.g., enzyme-linked immunosorbent assays) or liquid chromatography–mass spectrometry analytics, which can offer high sensitivity and specificity across a diverse range of protein targets[27,28]. More recently, another class of nucleic acid-based recognition elements, called aptamers, have been harnessed for protein detection and diagnostics[29,30]. Protein-binding aptamers are now available for specific binding to a wide variety of targets, including human proteins[31–34], HIV viral proteins[35,36], and bacterial toxins[37]. However, these assays require expensive detection reagents (e.g., purified antibodies or synthesized aptamers), expensive and bulky instruments, sample cold chain storage and distribution, and trained personnel. Instead, it is possible to utilize RNA-based aptamers to develop low-cost, genetically encoded riboswitch biosensors that carry out in situ protein detection within cell-free expression systems (TX-TL)[38]. Past efforts to engineer such riboswitch sensors have largely relied on trial-and-error experimentation, for example, constructing and characterizing large random libraries to identify riboswitch variants that work best. Instead, it is possible to apply biophysical modeling and computational design to engineer riboswitch sensors to directly couple protein binding to gene regulation, thereby creating a sense-and-respond capability without trial-and-error experimentation.

Here, we applied biophysical modeling and computational design to engineer protein-detecting riboswitches that directly regulate the

[1]Department of Agricultural and Biological Engineering, Pennsylvania State University, University Park, PA 16802, USA. [2]Department of Chemical Engineering, Pennsylvania State University, University Park, PA 16802, USA. [3]Department of Biomedical Engineering, Pennsylvania State University, University Park, PA 16802, USA. [4]Huck Institute Bioinformatics and Genomics Graduate Program, Pennsylvania State University, University Park, PA 16802, USA. ✉e-mail: salis@psu.edu

expression of a desired output protein within the TX-TL cell-free expression system, utilizing our Riboswitch Calculator algorithm to automatically convert RNA aptamers into designed riboswitch sequences[39]. We initially engineered riboswitches to detect the phage MS2 coat protein as a proof-of-principle, followed by engineering riboswitches to detect human monomeric C-reactive protein (mCRP) and interleukin-32 gamma (IL-32γ) as examples of medically relevant biomarkers. The best riboswitch sensors regulated reporter expression levels by 13.8, 15.9, and 2.5-fold when sensing the MS2, mCRP, and IL-32γ proteins, respectively, at biomarker concentrations of 1.25 μM mCRP and 0.78 μM IL-32γ. We demonstrated that these riboswitches controlled gene expression levels via two distinct mechanisms: (i) protein-induced conformational changes to RNA structure, which modifies the ribosome's ability to initiate translation; and (ii) protein-dependent steric repression, which blocks the ribosome from binding to the 5′ untranslated region. We critically tested the accuracy of the Riboswitch Calculator model predictions and found that improving the specification of the protein-aptamer interaction led to higher model accuracy. Overall, our automated design approach can be applied to convert any protein-binding RNA aptamer into a protein-detecting, cell-free biosensor with potential applications as portable, low-cost diagnostics.

## Results

### Riboswitch design and characterization platform

We created protein-binding riboswitch sequences using a biophysical model of translation-regulating riboswitches called the Riboswitch Calculator, which combines statistical thermodynamics with computational optimization to design synthetic riboswitches according to inputted specifications[39]. The design specifications include (i) the sequence of an RNA aptamer that binds to the protein of interest; (ii) the secondary structure of the RNA aptamer when it is bound by the

protein; (iii) the protein's binding free energy (or binding affinity) to the RNA aptamer; and (iv) the coding sequence of the protein whose expression level is regulated by the riboswitch (Fig. 1A). The Riboswitch Calculator then identifies synthetic pre-aptamer and post-aptamer sequences that maximize the riboswitch's dynamic range, utilizing a genetic algorithm to carry out computational multi-objective sequence optimization. Together, these pre-aptamer and post-aptamer sequences vary in length from 44 to 55 nucleotides, creating an overall searchable sequence space of $10^{26}$ to $10^{33}$ sequences. When designing riboswitches that activate translation (ON switches), the activation ratio is $R_{max} = \text{TIR}_{bound}/\text{TIR}_{unbound}$, where $\text{TIR}_{bound}$ and $\text{TIR}_{unbound}$ are the mRNA's translation initiation rates in the protein-bound and unbound states, respectively. When designing riboswitches that repress the translation rate (OFF switches), the repression ratio is $R_{max} = \text{TIR}_{unbound}/\text{TIR}_{bound}$. Multiple equally optimal riboswitch sequences are plausible. The Riboswitch Calculator identifies the Pareto-optimal set of synthetic riboswitch sequences that are all predicted to maximize $R_{max}$ with varying magnitudes of $\text{TIR}_{bound}$ and $\text{TIR}_{unbound}$.

To do this, the Riboswitch Calculator uses a statistical thermodynamic model, called the RBS Calculator, to calculate the interaction energies between the ribosome and mRNA that control its translation initiation rate[40,41]. The strengths of these interactions are determined by a 5-term Gibbs free energy model, including (i) conformational distortions when ribosomes initially bind to upstream standby sites in the mRNA; (ii) hybridization between the last 9 nucleotides of the 16S rRNA and the mRNA at the Shine–Dalgarno (SD) sequence; (iii) the unfolding of inhibitory mRNA structures that overlap with the ribosome's footprint, spanning the region from the 5′ end of the SD sequence to 13 nucleotides past the start codon; (iv) base pairing between the start codon and the initiating tRNA$^{fMet}$; and (v) ribosomal stretching or compression due to non-optimal spacer sequences

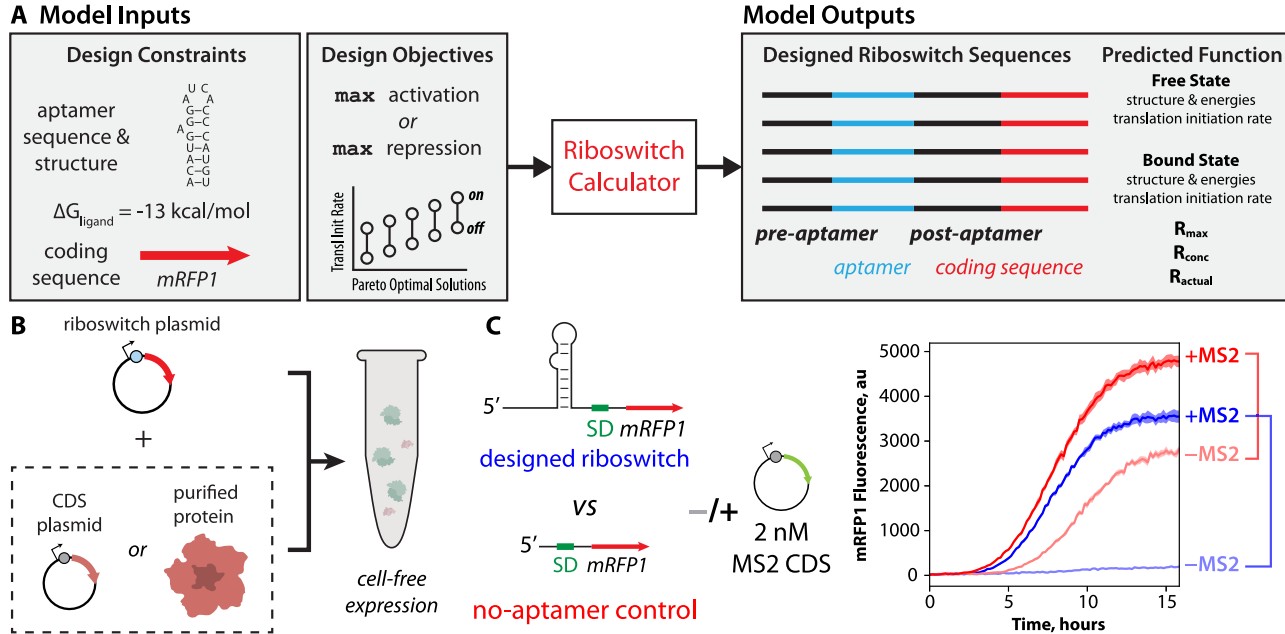

**Fig. 1 | Design of protein-sensing riboswitches and cell-free riboswitch characterization. A** A model of translation-regulating riboswitches takes as inputs the sequence, structure, and binding free energy of a protein-binding RNA aptamer. The model designs candidate riboswitch sequences that maximize activation or repression of translation initiation in response to changing protein concentrations. **B** The riboswitch sensor is tested in a cell-free expression system (TX-TL), adding either protein expression plasmid or purified protein. **C** Riboswitch sensor function is characterized by measuring reporter protein expression levels in response to changing protein-ligand concentrations alongside the same measurements on a no-aptamer control. The riboswitch activation or repression ratio is determined by comparing reporter expression levels, including the no-aptamer control measurements, to exclude non-specific interactions. Example cell-free assay measurements include (light blue), an MS2-sensing riboswitch without added MS2 protein; (dark blue) an MS2-sensing riboswitch with added MS2 expression plasmid; (light red) the no-aptamer control without added MS2 protein; and (dark red) the no-aptamer control with added MS2 expression plasmid. Lines are the mean mRFP1 fluorescence levels at each time point. Shaded regions are the 95% confidence interval at each time point ($N = 6$ biological replicates).

between the SD and start codon. The overall result is the binding free energy of the ribosome to the mRNA ($\Delta G_{total}$). By convention, a stronger interaction is denoted by a more negative binding free energy. According to Boltzmann's relationship, the mRNA's translation initiation rate is proportional to $\exp(-\beta\,\Delta G_{total})$, where beta is a constant that relates free energies to state probabilities[40].

To predict the riboswitch function, the Riboswitch Calculator carries out free energy calculations on the protein-bound and unbound states of the mRNA ($\Delta G_{total,bound}$ and $\Delta G_{total,unbound}$). In the unbound state, the RNA aptamer is indistinguishable from other parts of the mRNA and is allowed to form any secondary structure as part of the overall free energy minimization procedure, which utilizes the ViennaRNA suite of RNA folding algorithms[42]. In the protein-bound state, the RNA aptamer is locked into its protein-bound structure, which can alter the accessibility of the standby site, the folding free energies of inhibitory mRNA structures, and the overall binding free energy of the ribosome to the mRNA. The switching free energy is also calculated to determine the thermodynamic stability of the protein-bound state, using the protein's binding free energy to the RNA aptamer ($\Delta G_{ligand}$). The model predicts that the maximum fold-change in the mRNA's translation rate is $\exp(-\beta\,[\Delta G_{total,bound} - \Delta G_{total,unbound}])$ when the protein-bound state is stable and when an excess amount of protein is added. Riboswitch Calculator predictions were previously applied to engineer 62 synthetic riboswitches that detected a variety of small molecules (theophylline, tetramethylrosamineluoride, dopamine, thyroxine, 2,4-dinitrotoluene) and activated a protein reporter's translation rate by up to 383-fold[39]. However, the model has not yet been applied to convert protein-binding RNA aptamers into cell-free biosensors.

We characterized riboswitch function using the cell-free TX-TL expression system[43], adding plasmid-encoded genetic circuits that utilize each riboswitch to regulate the expression of the mRFP1 fluorescent protein reporter (Methods). We used two different approaches to add varying concentrations of protein ligands to the cell-free expression system: (i) co-expression or (ii) co-addition. For co-expression, we added varying amounts of a second expression plasmid to the cell-free expression system to constitutively produce the protein of interest. For co-addition, we directly added the purified protein of interest to the cell-free expression system. mRFP1 fluorescence levels were measured every 10 min using spectrophotometry (TECAN Spark), followed by endpoint analysis to quantify the overall change in reporter expression levels (Fig. 1B).

Prior work has demonstrated that gene expression in both the cellular and cell-free context is sensitive to added components or additional genetic load[44–49]. We carried out two types of controls to eliminate such confounding factors. In the first set of no-mRFP1 controls, we measured red fluorescence levels without any mRFP1 expression to quantify the autofluorescence of the cell-free assay and protein ligands. During endpoint analysis, we subtracted no-mRFP1 autofluorescence from all measurements. In the second set of no-aptamer controls, we measured red fluorescence levels from a genetic circuit that expresses mRFP1 using a standard 5′ UTR without any protein-binding aptamer (UTR-136[41]). According to the RBS Calculator v2.1 model, UTR-136 binds to the ribosome with a moderately high translation initiation rate (36,400 au or $\Delta G_{total} = -7.52$ kcal/mol). The purpose of this no-aptamer control is to measure any non-specific changes in red fluorescence levels when co-expressing or co-adding a protein-ligand at varying concentrations. If we detect non-specific activation or repression from the no-aptamer control, we remove this confounding factor by dividing the riboswitch's measured activation or repression ratio by the no-aptamer control's activation or repression ratio, respectively (Fig. 1C). If these riboswitch sensors are to be used as sensors in a future device, it would be expected that the same controls would be performed on the device, in parallel, to carry out these same measurements and analysis.

## Design and characterization of MS2, mCRP, and IL32γ Riboswitch Sensors

For the first proof-of-principle example, we engineered riboswitches to detect the phage MS2 coat protein, utilizing an RNA aptamer that folds into a well-defined hairpin structure and binds to MS2 protein with very high affinity ($K_d = 0.7$ nM)[50,51] (Fig. 2A). We then selected two medically relevant protein biomarkers, engineering riboswitches to detect changes in the levels of monomeric C-reactive protein (mCRP) and interleukin-32 gamma (IL-32γ). CRP is found in human plasma and natively forms a homopentameric complex, but will irreversibly dissociate into its monomeric form during a pro-inflammatory response (e.g., tissue damage, heart disease, cancer, or bacterial infection)[52,53]. An mCRP concentration of 10 mg/L (430 nM) is considered elevated, and depending on the disease severity, mCRP concentrations can readily exceed 100 mg/L (4.3 µM). To develop mCRP-sensing riboswitches, we harnessed a hairpin RNA aptamer that binds specifically to the monomeric form of CRP (mCRP) with high affinity ($K_d = 187.7$ nM) and does not bind to pentameric CRP[54]. IL-32 is a cytokine that regulates the NF-κB pathway and acts during early host defense against pathogen infections[55–57]. IL-32γ is the longest and most pro-inflammatory isoform of IL-32. Elevated levels of IL-32γ in human serum have been found to be associated with heart failure, COPD, and multiple myeloma[58] (up to around 61 pM or 1.6 ng/ml). To develop IL-32γ sensing riboswitches, we utilized a highly structured RNA aptamer that binds specifically to the gamma isoform with a high affinity ($K_d = 78$ nM)[59]. Altogether, we initially designed, constructed, and characterized 30 riboswitches to detect MS2, mCRP, and IL-32γ protein, including ON switches that activated mRFP1 expression and OFF switches that repressed mRFP1 expression (Fig. 2A). All sequences, model calculations, experimental and control measurements, and statistical significance tests (two-tailed $t$-tests with unequal variances) are available in the Source Data.

We first designed and tested five MS2-sensing riboswitches to activate mRFP1 expression (ON switches) and carried out TX-TL assays using the co-expression approach to produce the MS2 protein. When adding 8 nM pFTV1-MS2, we found that the designed MS2-sensing ON switches all increased mRFP1 fluorescence levels by significant amounts (between 10.8- to 27-fold). We carried out the same measurements using our no-aptamer control and found that co-expression of MS2 non-specifically increased mRFP1 fluorescence levels by 1.9-fold (Source Data). We then used the no-aptamer control measurement to remove the non-specific effect of MS2 on mRFP1 expression (Methods). After removing the non-specific effect, we found that all five MS2-sensing ON switches activated mRFP1 expression between 5.5- and 13.8-fold (Fig. 2B). These results show that the Riboswitch Calculator was able to design MS2-sensing ON switches that significantly activated output protein expression with the highest activated ratios reported to date, compared to prior efforts[26], though the dynamic ranges varied across the small number of designs tested.

We then designed and tested ten MS2-sensing riboswitches to repress mRFP1 expression (OFF switches), now adding 16 nM pFTV1-MS2 plasmid to the cell-free expression system as translation-repressing riboswitches are predicted to require higher protein concentrations to achieve a similar fold-change in output as compared to translation-activating riboswitches (Methods). Under the same conditions, the no-aptamer control increased mRFP1 fluorescence levels by 2.3-fold. In this scenario, a non-functional OFF switch would also cause mRFP1 fluorescence levels to increase by 2.3-fold due to this non-specific effect. Instead, we found that adding 16 nM pFTV1-MS2 plasmid caused all of the 10 MS2-sensing OFF switches to produce reduced mRFP1 fluorescence levels. After removing the non-specific effect, we found that their repression ratios varied from 2.5- to 5.3-fold (Fig. 2C). These results show that the Riboswitch Calculator can design MS2-sensing OFF switches by simply flipping its objective function during

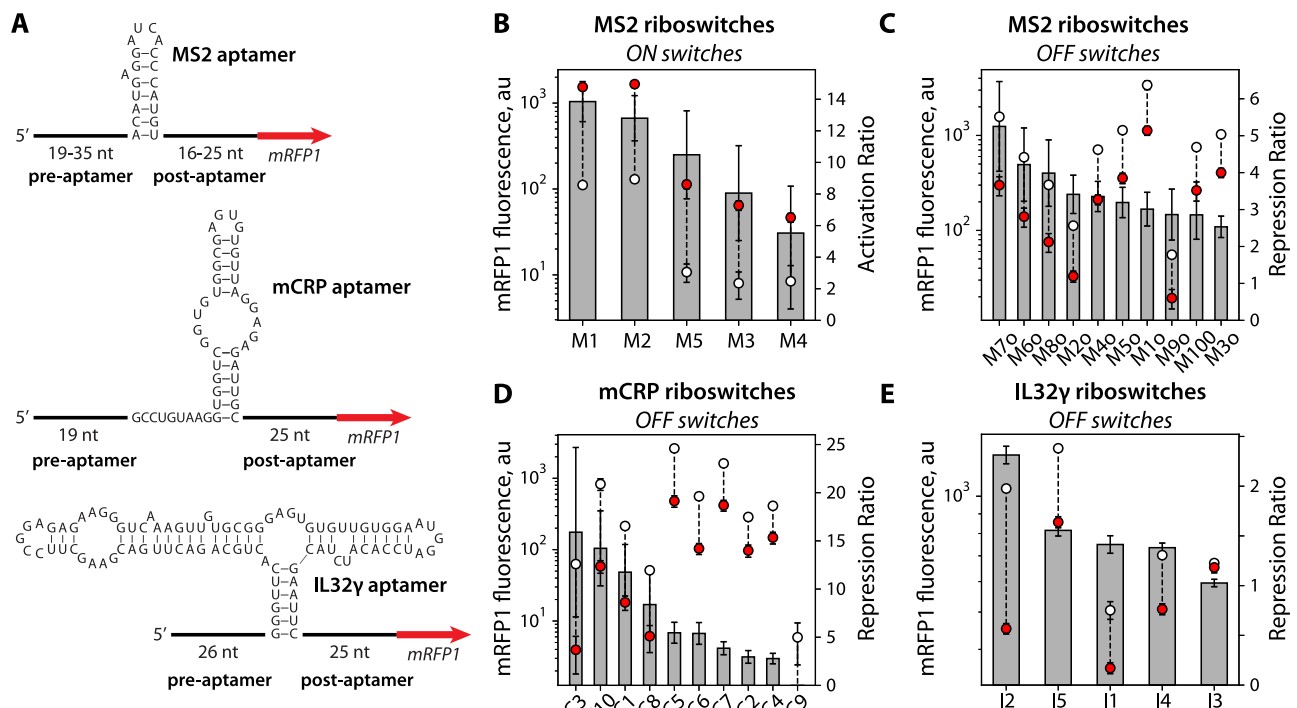

**Fig. 2 | Design and function of protein-sensing riboswitches. A** Riboswitch sequences designed for detecting the MS2, mCRP, and IL32γ protein ligands. **B** MS2 ON switches, induced with 8 nM MS2 CDS. **C** Characterization of MS2 OFF switches, induced with 16 nM MS2 CDS. **D** Characterization of mCRP OFF switches, induced with 1.25 μM of mCRP. **E** IL32γ OFF switches, induced with 780 nM IL32γ. Gray bars are the mean activation or repression ratios for designed riboswitches. White circles are mean mRFP1 fluorescence levels in the OFF state. Red circles are mean mRFP1 fluorescence levels in the ON state. Error bars are the 95% confidence intervals ($N$ = 6 biological replicates for MS2 ON switches; $N$ = 8 biological replicates for mCRP and IL32γ OFF switches).

sequence optimization while utilizing the same biophysical calculations.

We next designed ten mCRP-sensing riboswitches and characterized their ability to repress mRFP1 expression (OFF switches). We initially added 16 nM of a plasmid expressing mCRP (co-expression), but observed no significant change in mRFP1 regulation. Instead, when we directly added 1250 nM of purified mCRP (co-addition), we found that the OFF switches decreased mRFP1 fluorescence levels by up to 6.7-fold. As the purified mCRP was produced using HEK 293 cells, it is possible that chaperone-assisted protein folding or glycosylation is needed to produce an mCRP that binds to the RNA aptamer. Similar to the MS2 protein, we found from our no-aptamer controls that the addition of 1250 nM mCRP non-specifically activated mRFP1 expression by about 2.3-fold (Source Data). Once we removed this non-specific effect, we found that 90% of the mCRP-sensing OFF switches successfully repressed mRFP1 expression with repression ratios from 2.8-fold to 15.9-fold (Fig. 2D).

Finally, we designed five IL32γ-sensing riboswitches and characterized their ability to repress mRFP1 expression (OFF switches). We directly added 780 nM of purified IL32γ to the cell-free expression system (co-addition). Using our no-aptamer controls, we found no significant non-specific effect on mRFP1 expression levels (Source Data), precluding the need to remove any non-specific effect. After characterizing the OFF switches, we found that 4 out of the 5 riboswitches were able to significantly repress mRFP1 fluorescence levels from 1.4- to 2.5-fold (Fig. 2E). These results show that the Riboswitch Calculator was able to harness RNA aptamers that bind to human protein biomarkers to successfully design translation-repressing riboswitches.

### Structural and energetic contributions to riboswitch function

We next investigated the mechanisms responsible for riboswitch function to explain how protein binding to an aptamer domain can

cause the riboswitch's translation rate to be either activated or repressed. As examples, we focus on an MS2-binding ON switch (M2 riboswitch) and an MS2-binding OFF switch (M7o riboswitch), applying the Riboswitch Calculator model's structural and thermodynamic calculations to visualize and quantify the process[39]. In Fig. 3A, we show the sequence, structure, and interactions of the M2 riboswitch in its free state (state 1) and in its ribosome-bound state (state 2) in the absence of the MS2 protein. Before the ribosome has bound, the riboswitch mRNA is predicted to fold into a stable structure where the aptamer domain and SD sequence are both partly sequestered by base pairing. After the ribosome has bound to form a pre-initiation complex, there are significant structural re-arrangements, including unfolding an inhibitory structure within the N-terminal CDS region[40,60]. While the structures provide visual cues, the ribosome's ability to bind to the mRNA and initiate translation rate is actually controlled by the difference in Gibbs free energy between the initial and final states. In the absence of MS2, this difference is −3.7 kcal/mol.

We then illustrate how the mRNA's structure and ribosomal interactions are altered when the riboswitch is bound by MS2. When MS2 binds to its aptamer domain, the model predicts a substantial refolding of the 5' UTR, almost completely exposing the SD sequence and creating a highly accessible standby site for ribosome binding (Fig. 3A, state 3). This structural re-arrangement requires an input of at least 5.2 kcal/mol energy to push the transition forward, which is provided by the −13.2 kcal/mol energy released when MS2 binds to its aptamer domain. The MS2-bound mRNA can now bind to the ribosome to form a more stable pre-initiation complex (Fig. 3A, state 4) with a more negative binding free energy (−8.1 kcal/mol), leading to activation of the translation rate. Like all models, the calculations provide a simplified version of reality that nonetheless enable riboswitch prediction and design. For example, the model considers only the four most predominant states of the riboswitch, though there exists an ensemble of states with varied mRNA structures and MS2 binding

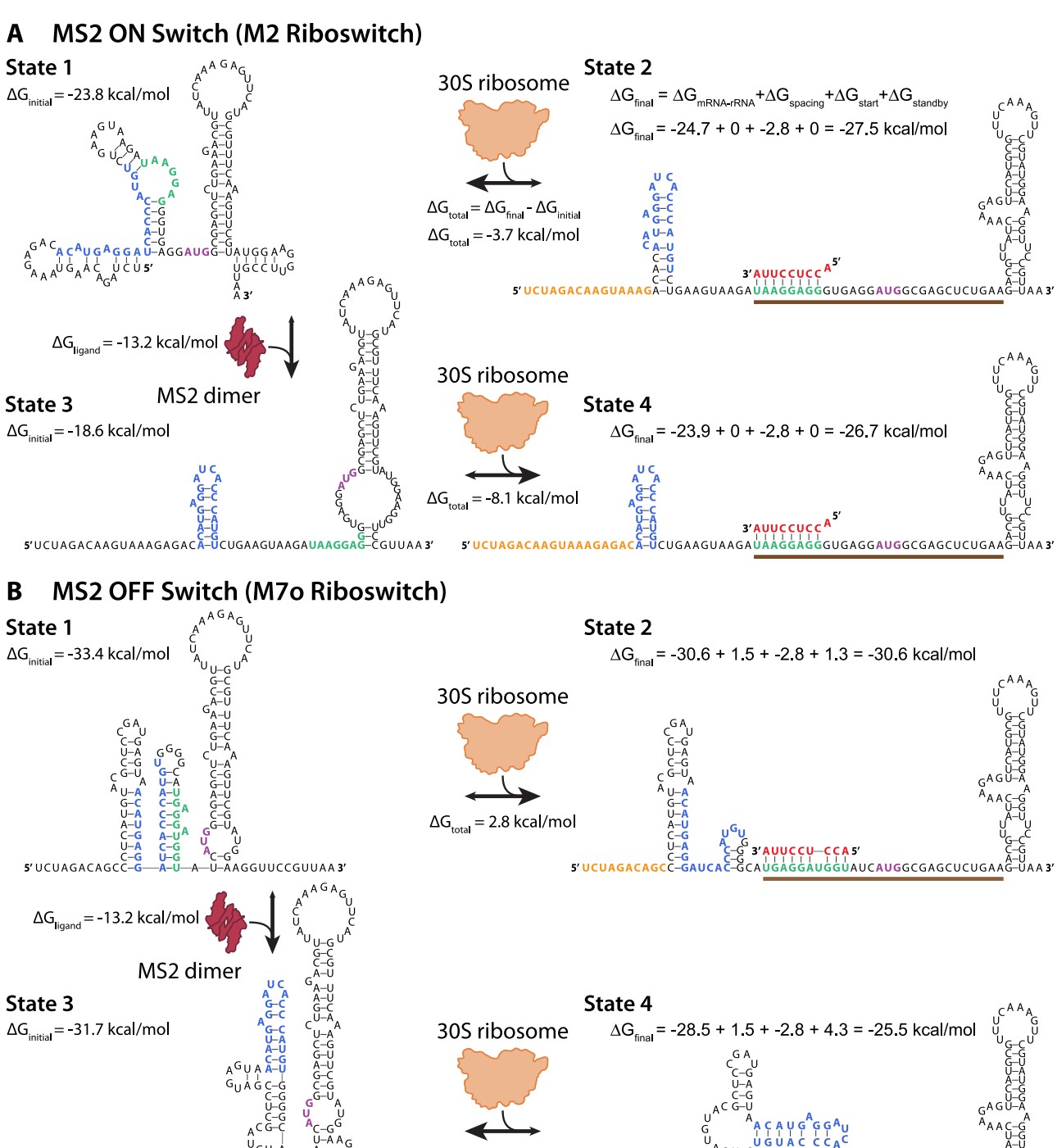

**Fig. 3 | Sequence, structure, and interactions controlling riboswitch function.** The model-predicted mRNA structures and ribosome-mRNA binding free energies for the **A** MS2 ON switch (M2 riboswitch) and the **B** MS2 OFF switch (M7o riboswitch) across its four most relevant states. State 1 shows the initial mRNA structure and its Gibbs free energy of folding ($\Delta G_{initial}$) when the MS2 coat protein is not bound. State 2 shows the change in mRNA structure and the final Gibbs free energy ($\Delta G_{final}$) when the ribosome binds to the mRNA in State 1. The free energy model for calculating $\Delta G_{final}$ includes hybridization between the mRNA and ribosomal RNA ($\Delta G_{mRNA\text{-}rRNA}$), base pairing between the tRNA and start codon ($\Delta G_{start}$), and energetic penalties for non-optimal spacing ($\Delta G_{spacing}$) and an inaccessible standby site

($\Delta G_{standby}$). State 3 shows how the mRNA structure changes when the MS2 coat protein binds to its cognate RNA aptamer with binding free energy $\Delta G_{ligand}$, which results in a change in mRNA folding free energy ($\Delta G_{initial}$). State 4 shows the change in mRNA structure and final Gibbs free energy ($\Delta G_{final}$) when the ribosome binds to the mRNA in State 3. The translation initiation rates are predicted based on the difference in initial and final Gibbs free energies ($\Delta G_{total}$) according to Boltzmann's relationship. Nucleotides are colored according to their interactions, including (blue) the RNA aptamer domain, (red) the last 9 nucleotides of the 16S ribosomal RNA, (green) the Shine–Dalgarno sequence, (purple) the start codon, and (orange) the standby site. The brown bar is the ribosomal footprint for initiation.

occupancies. Notably, MS2 binds to its aptamer while the mRNA is being transcribed, which can eliminate kinetic traps.

We next focus on the mechanism of an OFF switch, illustrating how MS2 binding triggers refolding of the mRNA to make it more energetically unfavorable for the ribosome to bind. In the absence of MS2, the model predicts that the initial state of the M7o riboswitch folds into highly stable mRNA structures that sequester the aptamer domain and the SD sequence (Fig. 3B, state 1). This mRNA structure does not favorably bind to the ribosome (2.8 kcal/mol), particularly due to the short distance between the SD and starts codon and the presence of mRNA structures that lower the accessibility of the standby site (Fig. 3B, state 2). However, once MS2 binds to the mRNA, the model predicts that the mRNA will bind even less favorably to the ribosome, due to a reconfiguring of both the initial and final mRNA states (Fig. 3B, states 3 and 4). The MS2-bound mRNA contains a highly stable structure that occludes the ribosome's standby site[41] and adds a penalty of 3 kcal/mol to the ribosome's total binding free energy (6.2 kcal/mol). Overall, using the model, we can quantitatively understand how MS2 binds to the aptamer domain and causes large-scale rearrangements in the riboswitch's mRNA structure to control ribosome binding free energies and translation initiation rates.

## Dose–response characterization of designed riboswitch sensors

We next selected high-functioning riboswitches (M2, C5, and I2) and measured how systematically increasing the MS2, mCRP, and IL32γ concentrations affected their output expression levels (dose–response) as compared to the no-aptamer control (UTR-136). All measurements and statistical tests are available in the Source Data. We first reconfirmed that the cell-free expression system had sufficient capacity to express large amounts of mRFP1 in proportion to the mRFP1 expression plasmid across a wide range (0–32 nM plasmid added) (Supplementary Fig. 1). We then characterized the dose–response of the M2 riboswitch by keeping the riboswitch plasmid concentration constant and systematically increasing the concentration of the MS2 expression plasmid from 0 to 16 nM. The M2 riboswitch activated mRFP1 fluorescence levels with a dose-dependent sigmoidal behavior, increasing mRFP1 expression levels by up to 26.6-fold (Fig. 4A). Under the same conditions, the mRFP1 fluorescence levels from the no-aptamer control increased by up to 1.86-fold. When correcting for this non-specific effect, the M5 riboswitch activated mRFP1 expression levels by 15.9-fold. Activation of mRFP1 expression was detectable with statistical significance ($p = 0.0008$, two-tailed $t$-test) using only 0.5 nM of expression plasmid. These results confirm the expected sigmoidal dose response for a translation-activating riboswitch, consistent with previously engineered riboswitches[39].

We then carried out the same dose–response characterization on the C5 riboswitch, directly adding up to 2500 nM of purified mCRP to cell-free assays. We found that the C5 riboswitch lowered mRFP1 fluorescence levels by 1.74-fold at 1250 nM mCRP and by 237-fold at 2500 nM mCRP (Fig. 4B). However, we found that the no-aptamer control exhibited a non-linear dosage response such that mRFP1 fluorescence levels were non-specifically increased by 2.37-fold at 1250 nM mCRP and non-specifically decreased by 11-fold at 2500 nM mCRP. Based on these measurements, mCRP non-specifically activates mRFP1 expression at lower concentrations, but then interferes with cell-free expression at higher concentrations. After taking into account these non-specific effects, the C5 riboswitch was found to repress mRFP1 expression by 4.1-fold at 1250 nM mCRP and by 21.5-fold at 2500 nM mCRP, though the interference with the cell-free assay at 2500 nM mCRP substantially increased the variability of the measurement. Repression was detectable with statistical significance at 156 nM mCRP, which is the lowest tested concentration above zero ($p = 0.0005$, two-tailed $t$-test), while half-maximal repression was achieved at 403 nM mCRP. These results show that the C5 riboswitch is sensitive enough to detect and quantify mCRP concentrations across

the physiological range from normal levels (100–430 nM) to elevated levels (above 430 nM), though levels above 2500 nM will inhibit cell-free expression.

We repeated the same cell-free assays using the I2 riboswitch, directly adding up to 780 nM of purified IL32γ. We found that the I2 riboswitch lowered mRFP1 fluorescence levels by 2.6-fold at the highest IL32γ concentration (Fig. 4C). In comparison, the no-aptamer control exhibited non-specific repression of only 1.09-fold, which was statistically indistinguishable from the baseline ($p = 0.18$, two-tailed $t$-test). Using the I2 riboswitch, repression of mRFP1 expression levels was detectable with statistical significance at a concentration of 195 nM IL32γ with half-maximal repression taking place at about 548 nM IL32γ ($p = 0.015$, two-tailed $t$-test). However, using the current IL32γ aptamer ($K_d = 78$ nM), the I2 riboswitch could not sense IL32γ levels within the picomolar range, which would be needed to distinguish between normal and elevated levels in human serum.

## Translation repression by steric hindrance and structural switching

We next investigated the mechanisms responsible for protein-induced translational repression with the goal of distinguishing between two types of interactions. When a protein binds specifically to the aptamer region of the mRNA, we anticipate that translational repression could be exerted by steric hindrance alone; a stably bound protein could prevent the 30S ribosomal subunit from associating with the mRNA, for example, at upstream standby sites, or it could block the 30S ribosomal subunit's 16S rRNA from hybridizing to the SD sequence to form a stable ternary complex. As a second mechanism, when a protein binds specifically to the aptamer region, it can induce changes in the mRNA structure, including the formation or removal of inhibitory mRNA structures that the ribosome must unfold prior to initiating translation. These mechanisms are layered on top of any non-specific interactions when adding protein to the cell-free system. Using a learn-by-design approach to distinguish these mechanisms, we designed and characterized several mCRP-binding riboswitches that employ either the first mechanism alone ("steric switches") or a combination of both mechanisms together ("OFF switches") alongside no-aptamer controls that measure the effects of non-specific interactions. For these experiments, we began using a second batch of purified mCRP that, through comparative testing on the same riboswitch (C10), exhibited about 4.1-fold less activity, though the cell-free composition remained the same (Source Data).

For the steric switch designs, we inserted the mCRP-binding aptamer upstream of a consensus SD sequence at varying distances (0–20 nt upstream of the 5' ends of the SD sequence). We also designed the pre-aptamer and post-aptamer so that the mRNA's structure remained the same in both the free and mCRP-bound states, eliminating the second mechanism as a source of translation regulation (Fig. 5A). We then characterized how well these riboswitches regulate mRFP1 expression levels, adding either 0 or 1250 nM mCRP. We found that placing the mCRP-binding aptamer directly upstream of the SD sequence lowered mRFP1 fluorescence levels by 3.2-fold. After using our no-aptamer controls to remove the effect of the non-specific interactions (1250 nM mCRP activated mRFP1 fluorescence levels by 1.65-fold), we found that specific binding of mCRP repressed mRFP1 expression by 5.3-fold, showing that steric hindrance alone is greatly contributing to protein-induced translational repression (Fig. 5B). When the aptamer was placed farther upstream of the SD sequence (5 or 20 nt), mRFP1 expression was repressed by smaller, but similar magnitudes, of 3.0 and 3.4-fold, respectively, again removing the non-specific interactions' contributions. These results show that steric hindrance can repress translation even when the aptamer is farther from the SD sequence.

In the second set of OFF switch designs, we applied the Riboswitch Calculator algorithm to design 5 additional mCRP-binding

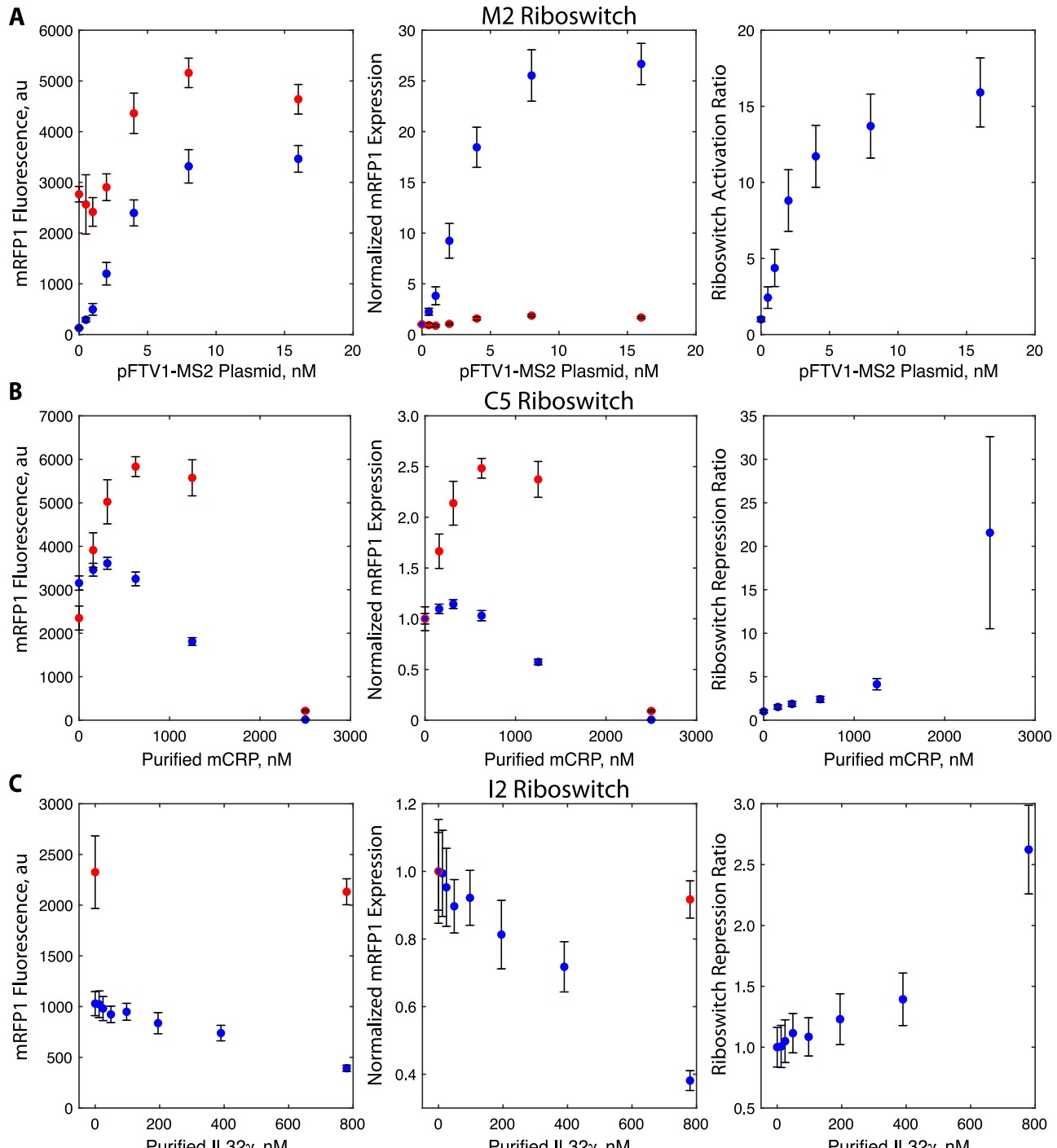

**Fig. 4 | Dose–response of MS2, CRP, and IL32γ riboswitches.** The measured mRFP1 fluorescence levels, normalized mRFP1 expression levels, and activation or repression ratios of the **A** M2, **B** C5, and **C** I2 riboswitch sensors (blue dots) in response to varied concentrations of the MS2 expression plasmid, purified human mCRP protein, and purified human IL32γ protein, respectively. The measured mRFP1 fluorescence levels and normalized mRFP1 expression levels of the no-aptamer control (UTR-136) under the same conditions (red dots). Dots and bars are the mean and standard deviation of replicate cell-free assays (**A** $N = 8$ biological replicates, **B** $N = 8$ biological replicates, **C** $N = 6$ biological replicates).

riboswitches (Co1 to Co5) while varying the distance between the aptamer region and the SD-like sequence from 4 to 13 nt. We found that mRFP1 expression levels were repressed by 3.2- to 13.6-fold (Fig. 5C) while taking into account the no-aptamer controls. Two of the OFF switches (Co1, Co2) had repression ratios far greater than the steric switches, showing that some of the designed riboswitches are utilizing both mechanisms to regulate translation rate. Overall, these results show that translation repression can take place by either steric

inhibition alone or through a combination of both steric inhibition and protein-induced structural re-arrangements that block ribosome binding.

**Accuracy analysis and improvement by modifying aptamer structural constraints**

We next critically analyzed the Riboswitch Calculator's model accuracy for protein-sensing riboswitches. We first considered the riboswitches'

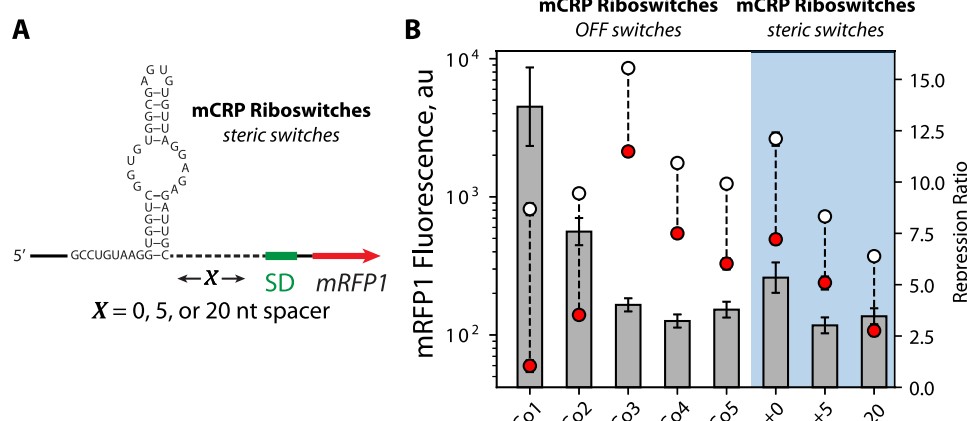

**Fig. 5 | Placement of mCRP aptamer in the standby site of the 5'UTR represses gene expression. A** The sequence and structure of the mCRP-binding steric switches, showing the spacer length ($X$) separating the mCRP aptamer and a consensus Shine–Dalgarno sequence. **B** The repression ratios of the designed mCRP OFF switches (Co1–Co5) and mCRP steric switches (+0, +5, +20) are compared. Gray bars are mean repression ratios. White circles are mean mRFP1 fluorescence levels in the OFF state. Red circles are the mean mRFP1 fluorescence levels in the ON state. Error bars are the 95% confidence intervals ($N = 8$ biological replicates).

OFF states where no protein-ligand is added. The Riboswitch Calculator predicts how well the ribosome binds to the mRNA as quantified by a change in binding free energy ($\Delta G_{total}$), which is expected to be related to the mRNA's translation initiation rate according to the log–linear Boltzmann relationship. We compared the riboswitches' model-predicted $\Delta G_{total}$ free energies to the natural logarithm of the end-point mRFP1 expression levels and found statistically significant and quantitative agreement ($R^2 = 0.47$, $p = 5.8 \times 10^{-6}$, $N = 34$) (Supplementary Fig. 2). However, we noticed that the expected dynamic range in translation rates was compacted in TX-TL assays as compared to equivalent in vivo assays. In other words, changing the ribosome's binding free energy to an mRNA had a smaller-than-expected effect on its translation rate. Quantitatively, the apparent Boltzmann factor for these TX-TL assays was $\beta = 0.23$ as compared to $\beta = 0.45$ in equivalent in vivo assays. This phenomenon has been previously observed in prior comparisons between cell-free and in vivo assays[49,61,62].

We then analyzed how well the Riboswitch Calculator predicted the activation or repression of mRFP1 expression levels, using three related calculations[39]: $R_{max}$, $R_{conc}$, and $R_{actual}$. $R_{max}$ is the maximum possible fold-change in translation rate when an excess amount of protein is added, and 100% of the riboswitch is bound by protein, while $R_{conc}$ is the fold-change in translation rate when a specified concentration of protein is added. $R_{actual}$ is an extension of the $R_{conc}$ calculation that additionally takes into account the thermodynamic stability of the riboswitch's protein-bound state; if the protein's binding free energy is insufficient to stabilize that state, the riboswitch can spontaneously transition back to its free state. The $R_{max}$ and $R_{conc}$ calculations make simplifying assumptions, whereas the $R_{actual}$ calculation is the complete version of the current model. In the first comparison, the $R_{max}$ prediction had a measurable but weaker correlation with the measured activation or repression ratios ($R^2 = 0.38$, $p = 9.3 \times 10^{-9}$, $N = 34$, linear regression) with only 15% of riboswitch variants regulating expression levels to within 2-fold of the predicted ratios (Fig. 6A). However, when we consider the proteins' concentrations using the $R_{conc}$ prediction, we find that the model more accurately predicted the riboswitches' activation or repression ratios ($R^2 = 0.46$, $p = 1 \times 10^{-5}$, $N = 34$, linear regression) with 38% of variants regulating expression to within 2-fold of the predicted ratio (Fig. 6B). Finally, using the $R_{actual}$ prediction to account for differences in the proteins' binding free energies to their respective RNA aptamers, the model accuracy increased even further ($R^2 = 0.48$, $p = 4.7 \times 10^{-6}$, $N = 34$, linear regression) with 44% of the riboswitch variants falling within the 2-fold tolerance (Fig. 6C).

A key input into the model calculations is the mRNA structure of the aptamer region when it is bound to the protein. When calculating the translation initiation rate of the riboswitch's protein-bound state, this mRNA structure is "locked" into place, which we call the aptamer structural constraint. In our initial model predictions, we extracted the aptamer region from the studies that developed it and assumed that the entire region is important to protein binding. However, it is possible that only a portion of the aptamer binds to its protein. For example, only 13 nucleotides of the 19-nt long MS2 aptamer are fully resolved in a crystal structure of the aptamer in complex with a MS2 coat protein dimer[50]. Considering that the mCRP and IL32γ aptamers have longer sequences (44-nt and 90-nt, respectively), it is likely that only a portion of these aptamers is truly "locked" into place when bound by protein.

We therefore investigated how changing the aptamer structural constraint alters the model's calculations and its predictive accuracy. To do this, we systematically varied the portion of the aptamer region that is "locked" in the protein-bound state, leaving a shorter contiguous region (Fig. 6D). For each of these shorter aptamer subregions, we recalculated its mRNA structure and used that structure as the aptamer's structural constraint. Here, any flanking unpaired nucleotides were left as unconstrained positions (dots) in the structural constraint, allowing them to refold with adjacent mRNA regions in the riboswitch's protein-bound state. We then repeated the model's $R_{actual}$ calculations. Throughout, all other riboswitch sequences and model parameters remain the same. Overall, changing the aptamer structural constraint varied the model's predicted $R_{actual}$ by up to 385-fold with a very rugged response surface, indicating that small changes to this input could have large effects on the model's predictions (Fig. 6D, Supplementary Fig. 3). We then identified the structural constraint for the mCRP and IL32γ aptamers that achieved the highest overall model accuracy across all riboswitches (Methods). Higher model accuracies were achieved by allowing more of the 5' portion of the mCRP aptamer to freely fold and by only "locking" a single bulged hairpin structure of the IL32γ aptamer. When inputting these aptamer structural constraints into the model (Source Data), we found that overall accuracy greatly improved ($R^2 = 0.64$, $p = 1.2 \times 10^{-8}$, $N = 34$, linear regression) with 56% of the riboswitch variants achieving less than 2-fold prediction error (Fig. 6E). Altogether, this analysis shows that accurately predicting riboswitch function requires correctly inputting the concentration of the protein-ligand, the binding free energy of the protein to its respective aptamer, and the actual aptamer structure that becomes "locked" when in its protein-bound state.

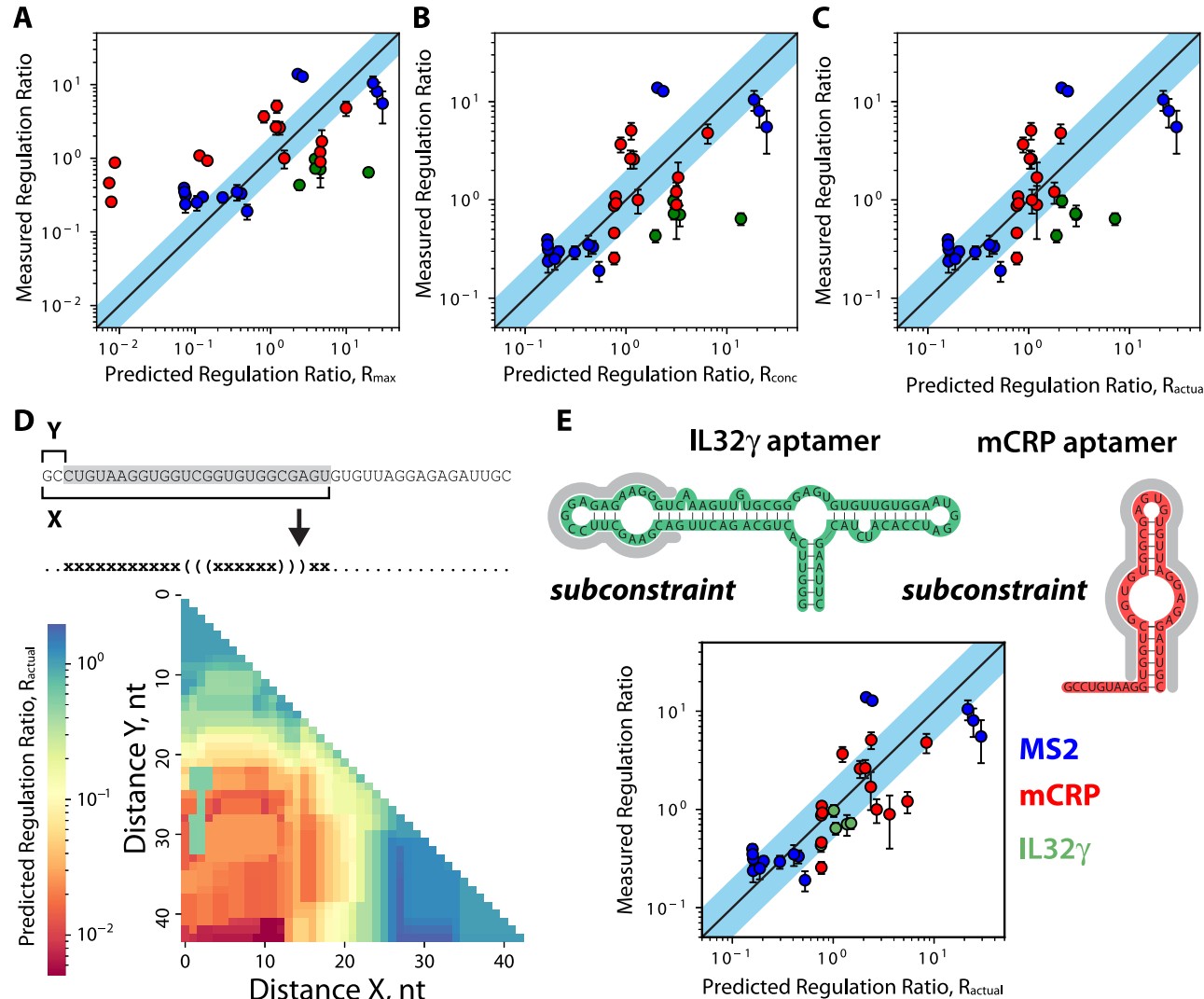

**Fig. 6 | Optimizing Riboswitch Calculator predictions for in vitro protein riboswitches.** Model predictions were compared to measured riboswitch regulation (activation or repression) ratios, utilizing three different equations that take into account additional interactions for improved accuracy. Model predictions include **A** the maximum predicted regulation ratio, $R_{max}$, **B** the predicted regulation ratio when considering the protein concentration, $R_{conc}$, and **C** the predicted actual regulation ratio, $R_{actual}$, when considering both the protein concentration and the thermodynamics of the mRNA-protein complex. **D** Model predictions were recalculated while systematically varying the length of the aptamer subconstraint region, showing riboswitch Co1 as an example. **E** The optimal aptamer subconstraints for the MS2, mCRP, and IL32γ aptamers were identified and utilized to predict the actual regulation ratio, $R_{actual}$, for all riboswitches. Overall accuracy is $R^2 = 0.64$, with 56% of riboswitches predicted to be within 2-fold of their measured activation ratio. Dots and error bars are the mean and standard deviation of data with 6–8 biological replicates.

## Discussion

We engineered 35 protein-sensing riboswitches that operate within cell-free expression assays, demonstrating an automated model-predictive workflow for converting protein-binding RNA aptamers into high-performance sensors with potential medical diagnostic applications. As demonstrations of our approach, we developed sensors for two human biomarkers—monomeric C-reactive protein and interleukin-32γ protein—that have been previously used as serum-accessible proxies for cardiovascular disease, pro-inflammatory conditions, and pathogen infection. As RNA aptamers can be developed to bind many proteins of interest, our design platform provides a reliable and versatile route to developing larger toolboxes of protein diagnostic assays using a variety of low-cost input and output formats.

We designed these genetically encoded protein sensors using a forward engineering approach, applying our Riboswitch Calculator algorithm to predict the riboswitches' sequence–structure–function relationship and computationally optimizing riboswitch sequences toward maximum translation activation or repression. Our use of computational design enabled us to reduce the large riboswitch sequence space to just a few designs that have a high chance of success. This approach yielded protein-sensing riboswitches with high activation or repression ratios—13.9-fold, 15.9, and 2.6-fold when detecting the MS2, mCRP, and IL-32γ proteins, respectively—at protein concentrations within the relevant physiological range. Notably, the biophysical model equations and parameters were fixed and constant throughout this study, demonstrating that a quantitative & predictive knowledge of biophysics from one realm can be applied to solve challenges in another realm without requiring new training or validation datasets.

We also investigated the key determinants of protein-sensing riboswitch function and identified two distinct mechanisms responsible for translation regulation. We began by engineering MS2-binding riboswitches that act as either ON and OFF switches, changing only their pre-aptamer and post-aptamer sequences, to show that protein-

induced changes to mRNA structure can be harnessed for either translation activation or repression. However, when designing mCRP-sensing OFF switches, we found that steric hindrance alone could explain how mCRP could bind, block the ribosome binding site, and repress the translation rate. While it was possible to engineer mCRP OFF switches that combined both steric inhibition and protein-induced structural inhibition at the same time to repress translation rate, we were not able to engineer mCRP ON switches after testing a small number of candidate designs. In practice, when developing a new protein-sensing riboswitch, it is worthwhile to design both ON and OFF switches to best determine how each protein interacts with the ribosome and whether steric inhibition plays an overriding role.

With our automated platform, it is now feasible to design large toolboxes of riboswitch sensors able to detect small molecule and protein ligands across a range of medical applications. For example, rather than using a fluorescent protein reporter, protein-sensing riboswitches can regulate the expression of enzymes, such as glucose oxidase, enabling electrical current generation as a measurable output signal[22]. Notably, cell-free expression systems have already been harnessed for such medical diagnostic applications. Though the addition of complex sample matrices (plasma, serum, urine, saliva) has been found to inhibit cell-free expression due to the presence of RNases, it has been shown that co-expression of RNase inhibitors can mitigate this inhibition, unlocking their potential[63]. A platform combining genetically encoded biosensors and cell-free expression could finally enable multiplexed protein detection in a compact, low-cost device that would provide longitudinal biomarker measurements in non-clinical settings for data-driven medicine.

## Methods

### Construction of riboswitch and protein-expressing plasmids

To construct riboswitch and CDS plasmids for this study, we started with the pFTV1 vector backbone, which contains mRFP1 modified to contain an N-terminal SacI restriction site[41]. We used the Riboswitch Calculator to design candidate riboswitch sequences and the Operon Calculator to codon-optimize the MS2 coat protein CDS and design an optimal RBS sequence (Source Data)[39,64]. We designed and ordered gBlocks, containing primer binding sites and additional restriction sites, and PCR primers for both the riboswitches and MS2 coat protein CDS (Integrated DNA Technologies). We PCR amplified the gBlocks using Phusion or Q5 DNA polymerase (New England Biolabs). For the riboswitches, we digested the riboswitch amplicons and pFTV1 vector backbone with XbaI and SacI-HF (New England Biolabs). For the MS2 coat protein CDS, we digested the CDS amplicon and pFTV1 vector backbone with XbaI and NotI-HF (New England Biolabs). For both the riboswitches and CDS, we ligated the digested inserts with digested backbone using T4 DNA ligase (New England Biolabs), and heat-shock transformed the ligation product into chemically competent DH10B. We then performed Sanger sequencing to verify that the insert had been cloned correctly.

### Crude cell lysate preparation

The crude cell lysate was prepared using the following protocol[43]. Twenty liters of *Escherichia coli* BL21 with the Rosetta2 plasmid encoding rare tRNAs were cultured in a Micros 30-l fermentor (New Brunswick) in 2XYT + P medium until the cells reached an OD600 of 1.5–2.0. The cell pellet was then collected in a T-1-P Laboratory continuous flow centrifuge (Sharples) and resuspended in 1 mL S30A buffer per gram of cell pellet. The resuspended cells were run through an M110-EH-30 microfluidizer (Microfluidics Corp.) at 20,000 PSI twice to ensure complete lysis. The lysate was clarified by centrifugation at $12,000 \times g$ for 30 min at 4 °C. The clarified lysate was then incubated for 80 min at 37 min while undergoing orbital shaking to perform the runoff reaction. After incubation, the lysate was centrifuged again at $12,000 \times g$ for 30 min at 4 °C. Following

lysis, clarification, and the runoff reaction, the lysate was diafiltered with a Pellicon Biomax 10 kDa MWCO 0.005 m² ultrafiltration module. Six retentate volumes of buffer S30B were run against the lysate at 4 °C. After diafiltration, the retentate was centrifuged for 30 min at $12,000 \times g$ at 4 °C. The protein concentration of the retentate was quantified using a Bradford BSA Protein Assay Kit assay (Bio-Rad). The retentate was aliquoted and flash-frozen in liquid nitrogen and stored at −80 °C.

### Cell-free expression reactions

Cell-free expression reactions were assembled on ice using the following protocol[43,46]. Amino acid and energy solutions were prepared separately and combined with crude cell extract to reach the following final concentrations: 7.4 mg/mL protein (1/3rd total reaction volume), 1.5 mM each amino acid (except for leucine at 1.25 mM), 50 mM HEPES, 1.5 mM ATP and GTP, 0.9 mM CTP and UTP, 0.2 mg/ml tRNA, 0.26 mM CoA, 0.33 mM NAD, 0.75 mM cAMP, 0.068 mM folinic acid, 1 mM putrescine, and 30 mM PEP. 4 mM additional magnesium glutamate (8.67 mM total), 80 mM additional potassium glutamate (100 mM total), and 2% w/v PEG-8000 were added to each reaction. Plasmid DNA was either miniprepped and ethanol precipitated or midiprepped and isopropanol precipitated and added to the reaction to a final concentration of 2 nM. Where protein was directly added to the reaction, mCRP (R&D Systems) or IL-32γ (Biotechne) was added at the specified concentration. Five microlitre reactions were incubated at 29 °C for 16 h in a 96-well polypropylene conical bottom plate sealed with a plate storage mat (Corning) in a TECAN Spark microplate reader. mRFP1 fluorescence was measured every 10 min, using 584 nm/607 nm ex/em with a 5 nm bandwidth.

### Endpoint mRFP1 data analysis

The endpoint was taken as the average of the last approximately 15 mRFP1 fluorescence data points of each reaction, during which active mRFP1 production had ceased. The following corrections were applied: the background for each reaction was calculated as the average of the first 15 data points, approximately, of each reaction, prior to the onset of mature mRFP1 accumulation. We observed that, even in the absence of plasmid DNA, there was a slight increase in the fluorescence between the beginning and end of each reaction, so each reaction was also corrected for the non-specific fluorescence increase by performing the same endpoint fluorescent calculation as above on a no-DNA reaction, and subtracting that non-specific fluorescence increase.

### Design of protein-detecting riboswitch sequences using the Riboswitch Calculator

We obtained the sequence and binding affinity for each aptamer used in this study from the literature. Aptamer secondary structures were determined using RNAfold (Vienna RNA v2.5), using the Turner 2004 nearest-neighbor parameter set, with no dangling end free energies[42]. To design the riboswitches, we used the design mode of the Riboswitch Calculator model of translation initiation regulation (Python v2.7 and v3.7.7), which builds on the RBS Calculator v2.1 model to predict the translation initiation states of each riboswitch in the uninduced and induced states[39,40]. From the output list of riboswitch sequences, we sub-selected sequences based on their predicted maximum regulation ratios and on- and off-state expression levels. Structural schematics were partly made using Forna diagrams[65].

### Alternate constraint analysis

To perform the alternate constraint analysis on the selected riboswitches, we iterated through every possible aptamer substring and re-calculated the predicted $R_{actual}$ for each substring. Briefly, for subconstraints i (first subconstrained nucleotide position) and j (last subconstrained nucleotide position), we re-folded the aptamer

subconstraint from position i to j using RNAfold as described above. We then appended the sequence from the beginning of the full aptamer constraint to i to the existing pre-aptamer sequence to generate a new pre-aptamer sequence and prepended the sequence from the end of the subconstraint to the end of the full aptamer sequence to the existing post-aptamer sequence to generate a new post-aptamer sequence. With the new pre-aptamer, post-aptamer, and aptamer sequences and structural constraint, we then calculated the predicted $R_{actual}$ at maximum induction using the Riboswitch Calculator.

## Statistical analysis
For pairwise comparisons, we used two-tailed, two-sample $t$-tests. For determining correlations, we used linear regression to calculate the Pearson squared correlation coefficient ($R^2$) and a hypothesis test to calculate the test statistic and $p$-value for the regression slope. All sample means standard deviations, replicate numbers, $t$-values, and $p$-values are provided in Source Data. For all tests, the significance level was set to $\alpha = 0.05$.

## Reporting summary
Further information on research design is available in the Nature Portfolio Reporting Summary linked to this article.

## Data availability
All sequences, model calculations, experimental & control measurements, and statistical significance tests are available in the Source Data. Source data are provided in this paper.

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

## Acknowledgements

This project was supported by funds from the Defense Advanced Research Projects Agency (HR001117C0095) and the Air Force Office of Scientific Research (FA9550-14-1-0089).

## Author contributions

G.E.V. and H.M.S. conceived the study, carried out designs and analyses, and wrote the paper. G.E.V. and L.R.G. carried out the experiments.

## Competing interests

A.U.S. patent application was filed on December 21, 2022, on protein-sensing riboswitches (application No. 63/434,212 assigned to Penn State University). H.M.S. is also a founder of De Novo DNA. G.E.V. and L.R.G. declare no competing interests.
