## [Peer Review File · Nature Communications]

Reviewers' Comments:

Reviewer #1:

Remarks to the Author:

The Authors of 'Automated Design of Protein-binding Riboswitches for Sensing Human Biomarkers in a Cell-free Expression System' describe how they applied their previously published 'Riboswitch Calculator' to create protein dependent RNA switches. They used three published protein binding RNA aptamers and fed them into their statistical thermodynamic model. The resulting sequences were placed upstream of the Shine-Dalgarno sequence of the coding sequence for the fluorescent protein mRFP1. They then tested the constructs in a cell-free TX-TL expression system, suggesting their use as biosensors for the biomarkers MS2, mCRP and IL32y.

While the authors claim to demonstrate how the 'Riboswitch Calculator' may be used for the creation of functional riboswitches from any aptamer, their work has conceptual problems and does not show the effectiveness of their model.

Major remarks:

The standardisation strategy used for all measurements, as described in lines 192-195 and Fig 1C, is insufficient. The authors used a no-aptamer control with and without added co-expression plasmid and divided the measured fluorescence ratios from the control ratios. This ratio is rather substantial, as shown in Fig 1C. It is likely, that the introduction of their aptamer-construct shifts the translation efficiency of co-expressed ligand proteins (MS2, mCRP, IL32y). The unspecific increase in measured mRFP1 fluorescence is likely not comparable to the used, external control. The observed, supposed riboswitching effect most likely stems from a steric-blocking effect of the ribosome by the bound protein. The authors claim to have observed a non-steric effect in lines 369-380. However, the presented data in Figure 4 do not support this claim. It is unclear what exactly is shown, as the Figure description does not seem to match the plots. Subfigure B is described to show 'mCRP off-switches, no-switch controls, and a mCRP 326 designed on-switch', but the data is labelled Co1-5 with no further explanation. Furthermore, a distance of 20 nucleotides (nt) is by far not enough to exclude a steric clash between ribosome and mCRP. 20 nt are about 85 Å in length, while the prokaryotic ribosome is about 200 Å and mCRP 40 Å across.

Therefore, the flanking sequences given by the 'Riboswitch Calculator' most likely are just not interfering with the aptamer structure. The conserved structure then binds the protein and sterically inhibits ribosome formation. The authors themselves write, that they only found weak correlation between their model and TX-TL measurements (lines 413-431) with R2 values from 0.38 to 0.48. The claim, that their model allows the creation of riboswitches from any aptamer does not hold true.

Lines 312-316, as well as the title, suggest a potential use as biosensors for human biomarkers. It is likely that the presented expression system will cease to function when tested with blood-serum, making them unfeasible for diagnostics.

In the discussion they write that 'RNA aptamers can be readily created to bind many proteins of interest' (475-476) and 'it is now feasible to design large toolboxes of riboswitch sensors able to detect small molecule and protein ligands across a range of medical applications (506-508). Both statements are incorrect as the selection of high quality RNA aptamers remains challenging, especially for small molecules which have not even been subject of their experiments.

Minor remarks:

The presented dose-response curves in Fig 3 show remarkable gaps that make it hard to determine any kinetics, especially with the sudden increases between the second to- and highest doses. This also leaves the high SD bars in subfigure B at 2500 nM somewhat unexplained. The authors have little regard for the difference between an 'aptamer' and a 'riboswitch', using both terms interchangeably throughout their manuscript. The authors speculate that post-translational modifications may be needed for mCRP, which are not present from co-expression in a TX-TL system. No data is shown to support this.

Reviewer #2:

Remarks to the Author:

In this manuscript Vezeau et al. present an automated design strategy of protein-binding riboswitches for sensing human biomarkers in a cell-free expression system. In 2016 the same lab published a very similar study using the same algorithm (Riboswitch Calculator) for the design of riboswitches to detect a variety of small molecules (Ref 37). The current manuscript applies Riboswitch Calculator to design protein-binding riboswitches.

In the first part the authors give a detailed explanation on how their Riboswitch Calculator works, followed by proof-of-concept studies using MS2, mCRP, and IL32 γ . In the second part they perform dose-response analysis of selected riboswitches and try to gain mechanistic insights into their mCRP OFF switches by using a learn-by-design approach (steric hindrance vs. structural switching). Lastly the authors critically investigate the Riboswitch Calculator's model accuracy.

Overall, the manuscript is well written and the data properly explained. However, there are some incidents where more in-depth explanation (supported by graphics) could improve the manuscript.

My concerns are mostly around their ON switch designs and their lack thereof for mCRP and IL32 γ . I think their MS2 ON switch is conceptually very interesting for the reader and deserves more attention than in the current version of the manuscript. Hence, I suggest minor revisions below.

Major Comments:

- In line 499 the authors state that it was surprising that the MS2 coat protein did not block ribosome binding. However, in Figure 2C they show MS2 OFF switches. Why the authors exclude a ribosome blocking mechanism for these sequences?
- How do the MS2 ON switches in Figure 2B work mechanistically? I suggest that the authors add an explanation to the main text and add an SI figure with predicted secondary structures of the OFF & ON state for one example (i.e., design M1 from Figure 2B).
- In line 502 the authors suggest to design and characterize both activating and repressing riboswitches. However, in this study they did not design and characterize ON switches for mCRP and IL32 γ . Could the authors please explain their reasoning? To me an ON switch would be more desirable for a diagnostic test since I would expect a reduced likelihood for false positives in a translation-ON detection system.

Minor Comments:

- In line 316 the authors argue that their IL32 γ switch is not sensitive enough to discriminate between normal and elevated levels of IL32 γ , which is expected since the K_d of the aptamer is too high to detect such low concentrations. Could the authors please explain why they chose the IL32 γ aptamer in the first place and not any other protein-binding aptamer with a more suitable K_d for the relevant physiological protein target concentration?
- It seems like that the data in Figure 4C is already included in Figure 4B. If this is the case, I would suggest to exclude Figure 4C from the manuscript to avoid redundancy.

Title: Automated Design of Protein-binding Riboswitches for Sensing Human Biomarkers in a Cell-free Expression System

Authors: Grace E. Vezeau, Lipika R. Gadila, Howard M. Salis

We would like to express our deep appreciation for the reviewers' constructive comments, which pointed out several areas where the manuscript could be improved. Below, in our point-by-point response, we answer the reviewers' questions and describe **additional figures** and **manuscript modifications** to further support our conclusions. The reviewers' questions are listed below (**black text**), followed by our responses (**blue text**). We have included a revised manuscript with highlighted changes as part of the peer-review information.

Reviewer #1:

The Authors of 'Automated Design of Protein-binding Riboswitches for Sensing Human Biomarkers in a Cell-free Expression System' describe how they applied their previously published 'Riboswitch Calculator' to create protein dependent RNA switches. They used three published protein binding RNA aptamers and fed them into their statistical thermodynamic model. The resulting sequences were placed upstream of the Shine-Dalgarno sequence of the coding sequence for the fluorescent protein mRFP1. They then tested the constructs in a cell-free TX-TL expression system, suggesting their use as biosensors for the biomarkers MS2, mCRP and IL32y. While the authors claim to demonstrate how the 'Riboswitch Calculator' may be used for the creation of functional riboswitches from any aptamer, their work has conceptual problems and does not show the effectiveness of their model.

We hope that the reviewers' suggested revisions to our manuscript as well as the clarifications below will resolve any concerns or ambiguities. For example, we would like to clarify a few points:

[1] The Riboswitch Calculator algorithm was responsible for designing the entirety of the sequence regions appearing before and after the RNA aptamer, which is between 44 to 55 nucleotides long altogether, encompassing a very large sequence space (10^{26} to 10^{33} possibilities). The post-aptamer sequence encompasses the Shine-Dalgarno (SD) sequence, making the SD part of the designed riboswitch sequence. Using only the thermodynamic model predictions, the algorithm identifies optimal nucleotide sequences to either maximally activate translation rates (ON switches) or maximally repress translation rates (OFF switches) when the RNA aptamer binds to the protein-of-interest. The Riboswitch Calculator uses a 5-term free energy model (the RBS Calculator) to calculate the ribosome's binding free energy to a mRNA and predict how that binding free energy is altered when a protein-of-interest binds to the embedded RNA aptamer. This change in binding free energy is then used during the sequence optimization procedure to design the riboswitches.

[2] We applied the Riboswitch Calculator to engineer 35 riboswitches utilizing 3 different protein-binding RNA aptamers. All 5 of the MS2 ON switches activated output protein expression levels from between 5.5 to 13.8-fold. All 10 of the MS2 OFF switches repressed output protein

expression levels from 2.9 to 5.3-fold. 9 out of 10 of the mCRP OFF switches repressed expression by 2.8 to 15.9-fold. 4 out of 5 of the IL-32 γ OFF switches repressed output protein expression by 1.4 to 2.3-fold. The model was highly effective at designing the 44-55 nucleotide sequence region surrounding the RNA aptamer to ensure that translation rates were activated or repressed when the protein of interest bound to the RNA aptamer (28 out of 30 successful designs). As the reviewer mentions, the model was more modestly effective at quantitatively predicting the activation/repression ratios for each of the individual designs. The model had a Pearson R^2 of 0.48 when forcing the entire RNA aptamer to fold into the structural shape suggested in the literature (shown in **Figure 2**). But if we allow the RNA aptamer to more freely fold with fewer structural constraints, then the model is more accurate with a Pearson R^2 of 0.64 (modified structures shown in **Figure 5**). Importantly, the Riboswitch Calculator model was never “trained” with any riboswitch characterization data; neither in this work nor in the original publication that described it. The free energy model itself is the same as when the Riboswitch Calculator was used to engineer riboswitches that bind small molecules & ions (e.g. theophylline, fluoride, dinitrotoluene) [Espah Borujeni, Amin, et al. "Automated physics-based design of synthetic riboswitches from diverse RNA aptamers." *Nucleic acids research* 44.1 (2016): 1-13.]. While there is room for improvement, the high rate of design success and a quantitative model accuracy of $R^2 = 0.48-0.64$ shows that the model's predictions are useful and valid when using it now to engineer protein-binding riboswitches.

[3] To our knowledge, this is the first study to design protein-binding riboswitches for human biomarker proteins, such as monomeric C-reactive protein and interleukin-32 gamma. Our key goal was to engineer riboswitches with sufficiently high activation/repression ratios to create a prototype low-cost assay for detecting & measuring proteins within a cell-free system. We clearly achieved that goal for our best mCRP OFF switches, and we showed that it's feasible to engineer an IL-32 γ OFF switch with a sufficiently high repression ratio to enable proxy measurements of IL-32 γ concentration within the tested range. It is perhaps not surprising that we see distinct riboswitch performances from the 3 proteins-of-interest as they have very distinct shapes & surfaces; MS2 and mCRP are globular whereas IL-32 γ has an extended shape with multiple disordered regions (**Figure 1**). These differences could impact steric inhibition of translation, though fully understanding how these differences contribute is outside the scope of the current study.

Response Figure 1: Measured or predicted structures of phage coat protein MS2, human monomeric C-reactive protein (mCRP), or human interleukin-32 gamma (IL-32 γ). Image scales vary.

Major remarks:

The standardisation strategy used for all measurements, as described in lines 192-195 and Fig 1C, is insufficient. The authors used a no-aptamer control with and without added co-expression plasmid and divided the measured fluorescence ratios from the control ratios. This ratio is rather substantial, as shown in Fig 1C. It is likely, that the introduction of their aptamer-construct shifts the translation efficiency of co-expressed ligand proteins (MS2, mCRP, IL32 γ). The unspecific increase in measured mRFP1 fluorescence is likely not comparable to the used, external control.

Yes, the reviewer is absolutely correct that the co-expression or co-addition of protein to the cell-free system can have a non-specific effect on protein expression levels, which is what our no-aptamer controls are measuring. The no-aptamer controls are normal 5' UTRs (ribosome binding sites) that express mRFP1 without any aptamer domain. We found that adding 8 nM of DNA template expressing MS2 protein increased the no-aptamer control's mRFP1 fluorescence from 2767 ± 152 to 5388 ± 434 units (a **1.94-fold** increase). In contrast, when we carried out the same experiment on our first MS2 ON switch (M1), its mRFP1 fluorescence increased from 111 ± 5 to 2999 ± 123 units (a **27-fold** increase). During our data analysis, we then removed the non-specific effect from the reported activation/repression ratios by dividing **27** by **1.94**, yielding the reported activation ratio of **13.8-fold**. This analysis was the same for all riboswitches. For the MS2 ON switches, the riboswitches have much lower mRFP1 fluorescences without MS2 present (8 to 130 fluorescence units) and much higher mRFP1 fluorescence with MS2 co-expressed (91 to 3237 fluorescence units). Without removing the non-specific effect, the increase in fluorescence varied from **10.7 to 27-fold**. When removing the non-specific effect (**1.94-fold** for MS2), the reported activation ratios were **5.5 to 13.8-fold**.

Second, to clarify, we added plasmid DNA to co-express the MS2 protein, but we directly added purified mCRP and IL32 γ to the cell-free system for detection. For the mCRP and IL32 γ OFF switches, we used the same procedure as above to measure and remove the non-specific effect. When we added 780 nM IL32 γ to the no-aptamer control, the mRFP1 fluorescence did not change in a statistically significant way (2325 ± 357 to 2132 ± 128) and there is no need to remove any non-specific effect. However, we found that adding 1.25 μ M mCRP altered mRFP1 fluorescence from 2197 ± 524 to 4663 ± 193 (a **2.1-fold** increase). The C10 mCRP OFF switch repressed mRFP1 fluorescence from 830 ± 223 to 124 ± 20 (a **6.7-fold** decrease). For OFF switches, when the non-specific effect increases protein expression levels, we need to convert the activation ratio into a repression ratio by simply taking its inverse. A 2.1-fold non-specific activation is the same as a 0.476-fold non-specific repression. We then remove the non-specific effect by dividing **6.7** by **0.476** to yield the reported repression ratio of **14-fold**. We carried out the same analysis on all mCRP OFF switches.

The reviewer may be concerned that removing the non-specific effect causes the repression ratio to increase for these mCRP OFF switches, but there is a sound logic to this procedure (which is the same procedure as for the ON switches). Our overall goal is to develop prototypes for a low-cost, protein-measuring, cell-free assay, while validating the use of our Riboswitch Calculator as a design tool. It is common in such assays to run parallel reactions where one reaction contains the recognition element and the other is a control. The assay outcome is always a quantitative comparison between the outputs from the two reactions. In glucose diagnostic devices, the measurement is a comparison between the current produced by a glucose oxidase enzyme versus a control current. Here, the outcome is a comparison of the fluorescent protein expression level from the riboswitch versus the no-aptamer control. Whether the control signal (the fold-change in fluorescence from the no-aptamer control) is above or below a value of one does not matter to the validity of the outcome. We should also note that our comparisons also eliminate the “arbitrary units” of fluorescence measurements; fold change activation or repression is unitless. We hope this clarifies our use of the no-aptamer controls and shows that – even if the non-specific effect is not removed from our measurements – our conclusions remain supported by our data (the M1 MS2 ON switch increases mRFP1 fluorescence by 27-fold and the C10 mCRP OFF switch decreases mRFP1 fluorescence by 6.7-fold).

We have updated our manuscript text to better describe how the no-aptamer control measurements are used to remove the non-specific effect from the reported activation and repression ratios.

The observed, supposed riboswitching effect most likely stems from a steric-blocking effect of the ribosome by the bound protein. The authors claim to have observed a non-steric effect in lines 369-380. However, the presented data in Figure 4 do not support this claim. It is unclear what exactly is shown, as the Figure description does not seem to match the plots. Subfigure B is described to show ‘mCRP off-switches, no-switch controls, and a mCRP designed on-switch’, but the data is labelled Co1-5 with no further explanation. Furthermore, a distance of 20 nucleotides (nt) is by far not enough to exclude a steric clash between ribosome and mCRP. 20 nt are about 85 Å in length, while the prokaryotic ribosome is about 200 Å and mCRP 40 Å across. Therefore, the flanking sequences given by the ‘Riboswitch Calculator’ most likely are just not interfering with the aptamer structure. The conserved structure then binds the protein and sterically inhibits ribosome formation. The authors themselves write, that they only found weak correlation between their model and TX-TL measurements (lines 413-431) with R2 values from 0.38 to 0.48. The claim, that their model allows the creation of riboswitches from any aptamer does not hold true.

Yes, the reviewer is absolutely correct that the steric effect is an important contributor to the translation repression observed in the mCRP OFF switches. However, our dataset shows that there is a non-steric contribution for some of the mCRP OFF switches (Co1 and Co2 designs). We agree that the description of our results in this section was not clear and needed to be greatly improved. To clarify, we designed and tested three “steric switches” that contain an RNA aptamer and a consensus Shine-Dalgarno sequence, separated by a spacer with varying length (between 0 to 20 nucleotides). These steric switches are labeled +0, +5, and +20 for their spacer length. We

then designed and tested five new mCRP OFF switches where we systematically increased spacer length between the RNA aptamer and the SD sequence within the post-aptamer region (+4 to +13 nucleotides). The measured repression ratios for the steric switches varied from 5-fold for the +0 spacer length to about 3-fold for the +5 and +20 spacer lengths. These results showed that steric inhibition can repress translation, but with modest repression ratios of 3 to 5-fold. The measured repression ratios for the new mCRP OFF switches varied from 3.2 to 13.7-fold with two of the designs (Co1 and Co2) having repression ratios (13.7 and 7.6-fold) that were significantly higher than the steric switches (3 to 5 fold). Steric inhibition alone cannot explain how the Co1 mCRP OFF switch is able to repress mRFP1 expression by 13.7-fold, suggesting that the mechanism involves both steric inhibition as well as structural switching. Our objective with these experiments was to develop steric-only switches to use as a comparison against the OFF switches to quantify the differences. We clearly see that some mCRP OFF switches (Co3, Co4, and Co5) have repression ratios that could be explained by solely steric inhibition whereas others (Co1, Co2) have higher repression ratios that require a combination of both mechanisms. We have updated our manuscript text to better describe the results in this section.

We would also like to clarify that the steric effect is likely to be protein specific. As we have shown, the MS2 protein is able to bind to its RNA aptamer and activate translation rates by 13.8-fold via ligand-induced structural switching, which improves ribosome binding. The MS2 protein is 130 amino acids long and forms a dimer (27.4 kDa) with a 33 Å end-to-end distance.

Regarding our model accuracy & versatility, as described above, the model was highly effective at designing protein-sensing riboswitches (determining the 44-55 nucleotide sequence region surrounding the RNA aptamers) so that translation rates were activated or repressed in response to protein binding. Out of 30 riboswitch designs, 28 achieved their target function, even though the designed sequence regions & aptamer structures are quite distinct. As the reviewer mentions, the model was more modestly effective at quantitatively predicting the activation/repression ratios for each of the individual designs. But, to clarify, the Pearson R^2 of 0.38 only occurs when we purposefully ignore a set of interactions in the model to see how they affect the prediction. The complete model has a Pearson R^2 of 0.48 when folding the entire RNA aptamer, which is improved to a Pearson R^2 of 0.64 when only allowing a portion of the RNA aptamer to fold. These are not weak correlations, particularly considering that these are “one shot” predictions with a small number of trials/experiments. A brief reminder that a Pearson R^2 of 0.64 is equivalent to a Pearson R of 0.80. Some studies only report correlation in terms of R.

Lines 312-316, as well as the title, suggest a potential use as biosensors for human biomarkers. It is likely that the presented expression system will cease to function when tested with blood-serum, making them unfeasible for diagnostics.

Yes, the addition of human sample matrices (plasma, serum, urine, saliva) inhibits protein expression levels in cell-free assays, mainly due to the presence of RNases. However, significant prior work has been carried out to overcome this challenge. A recent study [Voyvodic, Peter L., et al. "Evaluating and mitigating clinical samples matrix effects on TX-TL cell-free performance." *Scientific Reports* 12.1 (2022): 1-9.] found that co-expression of RNase inhibitors inside the cell-

free assay greatly increases protein expression levels. This is a superb & low-cost solution as it only requires the addition of a plasmid template, just like our protein-sensing riboswitches. With such solutions, it is feasible to use our protein-sensing riboswitches as a diagnostic. We have added this recent development to our Discussion section, including a citation to this study.

In the discussion they write that ‘RNA aptamers can be readily created to bind many proteins of interest’ (475-476) and ‘it is now feasible to design large toolboxes of riboswitch sensors able to detect small molecule and protein ligands across a range of medical applications (506-508). Both statements are incorrect as the selection of high quality RNA aptamers remains challenging, especially for small molecules which have not even been subject of their experiments.

Yes, we agree that our wording does not correctly communicate the difficulty of developing high quality RNA aptamers. We have modified the manuscript text accordingly. However, since the development of the SELEX process, there are now hundreds of aptamers that bind to diverse targets with varying affinities (“quality”), including everything from ions and small molecules to large proteins. The key purpose of the Riboswitch Calculator is to provide a reliable way to convert high quality aptamers into genetically encoded translation-regulating riboswitch sensors. In our prior work, we applied the Riboswitch Calculator to engineer riboswitches that sense 6 different small molecules and ions (theophylline, tetramethylrosamine, dopamine, thyroxine, fluoride, and 2,4-dinitrotoluene) utilizing 6 different RNA aptamers [Espah Borujeni, Amin, et al. "Automated physics-based design of synthetic riboswitches from diverse RNA aptamers." *Nucleic acids research* 44.1 (2016): 1-13.] However, we had never used the Riboswitch Calculator to engineer protein-sensing riboswitches, which is the topic of this work.

Minor remarks:

The presented dose-response curves in Fig 3 show remarkable gaps that make it hard to determine any kinetics, especially with the sudden increases between the second to- and highest doses. This also leaves the high SD bars in subfigure B at 2500 nM somewhat unexplained.

Yes, we carried out the dose response measurements using logarithmically-spaced concentrations of the MS2-expressing plasmid, purified mCRP, and purified IL32γ. However, the data in Figure 3 is shown with a linearly scaled x-axis. We’ve created versions of these plots using a logarithmically-scaled x-axis, but the perceptive difference when examining the plot is not large. There is also a benefit to clearly showing the protein concentrations used when determining the dose response, which is better achieved when using a linearly scaled x-axis, even if the plot doesn’t “look as good”. The large standard deviation in repression ratio at 2500 nM mCRP is due to the fact that a standard deviation of a ratio depends on the standard deviation of both the numerator and denominator. Here, when 2500 nM mCRP is added to the cell-free system, the mCRP-sensing ribowitch reduces mRFP1 fluorescence to a very low number, but that low number has a high coefficient of variation (std dev/mean), which causes the repression ratio to have a high standard deviation. This is fully expected in this scenario.

The authors have little regard for the difference between an ‘aptamer’ and a ‘riboswitch’, using both terms interchangeably throughout their manuscript.

We have also adjusted our text to ensure correct usage of the words “aptamer” and “riboswitch”. The aptamer is the portion of the nucleic acid that binds to the target analyte whereas the riboswitch is the longer portion encompassing the entire 5’ untranslated region (5’ UTR). This is consistent with terminology used in the field.

The authors speculate that post-translational modifications may be needed for mCRP, which are not present from co-expression in a TX-TL system. No data is shown to support this.

We have modified the manuscript text to avoid speculating as to the reason why co-expression of mCRP does not cause riboswitch regulation in contrast to co-addition of purified mCRP, which does cause riboswitch regulation. We do provide the facts that (i) the purified mCRP was produced using HEK 293 cells and (ii) HEK 293 cells carry out chaperone-assisted protein folding and glycosylation, unlike the cell-free assay.

Reviewer #2 (Remarks to the Author):

In this manuscript Vezeau et al. present an automated design strategy of protein-binding riboswitches for sensing human biomarkers in a cell-free expression system. In 2016 the same lab published a very similar study using the same algorithm (Riboswitch Calculator) for the design of riboswitches to detect a variety of small molecules (Ref 37). The current manuscript applies Riboswitch Calculator to design protein-binding riboswitches. In the first part the authors give a detailed explanation on how their Riboswitch Calculator works, followed by proof-of-concept studies using MS2, mCRP, and IL32 γ . In the second part they perform dose-response analysis of selected riboswitches and try to gain mechanistic insights into their mCRP OFF switches by using a learn-by-design approach (steric hindrance vs. structural switching). Lastly the authors critically investigate the Riboswitch Calculator’s model accuracy. Overall, the manuscript is well written and the data properly explained. However, there are some incidents where more in-depth explanation (supported by graphics) could improve the manuscript.

My concerns are mostly around their ON switch designs and their lack thereof for mCRP and IL32 γ . I think their MS2 ON switch is conceptually very interesting for the reader and deserves more attention than in the current version of the manuscript. Hence, I suggest minor revisions below.

We thank the reviewer for their constructive and insightful comments! We’ve modified the manuscript text and included more illustrative schematics to address their concerns, which have greatly improved the manuscript.

Major Comments:

- In line 499 the authors state that it was surprising that the MS2 coat protein did not block ribosome binding. However, in Figure 2C they show MS2 OFF switches. Why the authors exclude a ribosome blocking mechanism for these sequences?

Yes, we show that is possible to engineer MS2-binding riboswitches that either activate or repress translation, changing only the pre-aptamer and post-aptamer sequences surrounding the aptamer domain. We do not exclude a ribosome blocking mechanism for the MS2 OFF switches. We apologize for the ambiguity here and have modified the Discussion and Results sections to clarify. As the reviewer suggested, we've included illustrative schematics showing how the MS2 ON and OFF switches work.

In the case of the MS2 OFF switches, the MS2 aptamer needs to be unfolded by the ribosome to initiate translation and MS2 binding to the aptamer will make that unfolding process require a larger energetic input, which lowers the mRNA's translation rate. However, by changing the MS2 aptamer's location (with respect to the Shine-Dalgarno sequence), we can position it to be either within or outside of the ribosome's binding footprint. If the aptamer domain is outside the ribosomal footprint, it no longer needs to be unfolded by the ribosome, and it's now possible to design MS2 ON switches that activate translation rate (by designing the pre-aptamer & post-aptamer sequences so that MS2 binding causes unfolding of other inhibitory mRNA structures). All of these design decisions are in fact autonomously made during the sequence optimization procedure, using the Riboswitch Calculator model predictions to predict how MS2 binding will alter the mRNA's structure in the free and MS2-bound states.

- How do the MS2 ON switches in Figure 2B work mechanistically? I suggest that the authors add an explanation to the main text and add an SI figure with predicted secondary structures of the OFF & ON state for one example (i.e., design M1 from Figure 2B).

Yes, this is a great suggestion. We have added a new section to our Results and have added a new **Figure 3** to illustrate how the MS2 ON and OFF switches work mechanistically. The schematic shows the sequences & secondary structures for each of the riboswitch's four states: the free mRNA, the mRNA bound by the 30S ribosomal subunit, the MS2-bound mRNA, and the MS2-bound mRNA also bound by the 30S ribosomal subunit. For each state, we show the thermodynamic calculations (Gibbs free energies) and how these calculations are combined to determine the ribosome's binding free energy to the mRNA. In the text, we use the schematic to explain how MS2 binding to its aptamer causes structural re-arrangements in the mRNA that lead to either higher translation rates (more negative ribosome binding free energies) for the ON switch or lower translation rates (more positive ribosome binding free energies) for the OFF switch. For the OFF switch, the structural re-arrangements also affect the accessibility of the standby site, which appears upstream of the Shine-Dalgarno sequence.

- In line 502 the authors suggest to design and characterize both activating and repressing riboswitches. However, in this study they did not design and characterize ON switches for mCRP and IL32γ. Could the authors please explain their reasoning? To me an ON switch would

be more desirable for a diagnostic test since I would expect a reduced likelihood for false positives in a translation-ON detection system.

Yes, we attempted to engineer translation-activating riboswitches (ON switches) for mCRP and IL32 γ , but did not find functional ones. For IL32 γ -sensing ON switches, we observed either no change in expression or a slight repression of expression. For mCRP-sensing ON switches, we observed a strong repression of expression. These results motivated us to investigate the steric inhibition of translation rate by mCRP binding by designing & testing the mCRP-binding steric switches. We found that mCRP strongly represses translation rate, even though the mCRP aptamer was positioned similarly to the aptamer in an MS2 ON switch, suggesting that mCRP can block ribosome binding even when the aptamer is positioned outside the ribosomal footprint. This would also explain why engineering mCRP ON switches was not feasible (at least within the small number of designs that were tested). In general, when utilizing OFF switches within a diagnostic device, it would be necessary to run several types of controls in parallel to prevent the occurrence of false positives (e.g. no-aptamer controls and no-sample controls). However, to prevent the occurrence of false negatives, it would also be important to run similar controls when utilizing ON switches. Regardless of the switch mode, controls need to be run in parallel to ensure switch performance.

Minor Comments:

- In line 316 the authors argue that their IL32 γ switch is not sensitive enough to discriminate between normal and elevated levels of IL32 γ , which is expected since the K_D of the aptamer is too high to detect such low concentrations. Could the authors please explain why they chose the IL32 γ aptamer in the first place and not any other protein-binding aptamer with a more suitable K_D for the relevant physiological protein target concentration?

When the study was initiated, our major question was whether the Riboswitch Calculator could be applied to design protein-sensing riboswitches. As this is the first study to engineer a riboswitch that binds human proteins, it was not clear which human proteins should be selected. We selected protein-binding aptamers based on their reported affinities and their structural complexity (lower K_D and structural complexity were ranked higher). Our main focus was developing & applying a workflow to engineer these riboswitches and identify their mechanisms. Once we found that the design algorithm was working as expected, we then investigated the physiological concentrations of the proteins in serum across normal and diseases conditions. The costs of purified protein reagents were also a factor in selecting proteins of interest, which is important when testing riboswitch function across many replicates.

- It seems like that the data in Figure 4C is already included in Figure 4B. If this is the case, I would suggest to exclude Figure 4C from the manuscript to avoid redundancy.

Yes, we removed Figure 4C and clarified in the legend that the +0, +5, and +20 are the spacer lengths.

Reviewers' Comments:

Reviewer #1:

Remarks to the Author:

While the authors have addressed many of the raised concerns and have made notable improvements to their manuscript, one fundamental problem persists and has been confirmed with the explanations given by the authors in the rebuttal.

The normalization strategy employed for the riboswitch measurements is invalid and greatly distorts the presented results. The authors calculate the mRFP1 fluorescence ratio of their riboswitch construct with and without the respective protein present. They then divide it by the ratio of their 'no aptamer' control with and without protein. This method of normalization cannot truthfully represent the underlying principles of their measurement.

The addition of the respective protein will have mostly (if not fully) additive or subtractive impact on the measured values and no multiplicative effects. One expects absorption, autofluorescence of the respective protein and the likes. The change in fluorescence observed by protein (or plasmid for translation) addition must therefore be subtracted from the fluorescence of the riboswitch construct with added protein, before calculating the ratio: $(FLR(\text{riboswitch w/ protein}) - \Delta FLR(\text{no-aptamer})) / FLR(\text{riboswitch w/o protein})$. For example, in case of the calculation for MS2 – as described in the rebuttal – that gives a real regulation of 3.4-fold, not 13.8-fold as the authors claim.

Reviewer #2:

Remarks to the Author:

The authors have addressed the key concerns raised in the original round of review in a satisfactory manner.

Title: Automated Design of Protein-binding Riboswitches for Sensing Human Biomarkers in a Cell-free Expression System

Authors: Grace E. Vezeau, Lipika R. Gadila, Howard M. Salis

We would like to express our gratitude for the reviewers' constructive questions and supportive comments. Below, in our point-by-point response, we describe results from **recent experiments** and **first-principles equations** to directly answer the reviewers' questions. The reviewers' questions are listed below (**black text**), followed by our responses (**blue text**). We have updated our Supplementary Data to include the data from the new experiments.

Reviewer #1:

While the authors have addressed many of the raised concerns and have made notable improvements to their manuscript, one fundamental problem persists and has been confirmed with the explanations given by the authors in the rebuttal. The authors calculate the mRFP1 fluorescence ratio of their riboswitch construct with and without the respective protein present. They then divide it by the ratio of their 'no aptamer' control with and without protein. This method of normalization cannot truthfully represent the underlying principles of their measurement. The addition of the respective protein will have mostly (if not fully) additive or subtractive impact on the measured values and no multiplicative effects. One expects absorption, autofluorescence of the respective protein and the likes. The change in fluorescence observed by protein (or plasmid for translation) addition must therefore be subtracted from the fluorescence of the riboswitch construct with added protein, before calculating the ratio: $(FLR(\text{riboswitch w/ protein}) - \Delta FLR(\text{no-aptamer})) / FLR(\text{riboswitch w/o protein})$. For example, in case of the calculation for MS2 – as described in the rebuttal – that gives a real regulation of 3.4-fold, not 13.8-fold as the authors claim.

Response Figure 1: We carried out kinetic spectrophotometry measurements on cell-free assays to measure the auto-fluorescence of MS2 protein. Adding the MS2 expression plasmid to a cell-free assay does not increase the fluorescence levels (575 / 620 nm) as compared to a cell-free assay without any plasmid template.

The reviewer raises two very interesting and distinct questions, which we can directly answer. The first question is whether the MS2 protein emits auto-fluorescence at the excitation & emission wavelengths used to measure mRFP1 expression levels. If the MS2 protein emits auto-fluorescence, then we agree that we would need to subtract this auto-fluorescence from our measurements. To answer this question, we measured the kinetics of fluorescence levels when either (a) a cell-free assay is carried out without adding any plasmid template or (b) a cell-free assay is carried out when adding the MS2 expression plasmid. In

both cases, no plasmid is added that expresses the mRFP1 reporter protein. We carried out these fluorescence measurements in the same way as our riboswitch characterization experiments (kinetic spectrophotometry over a 12-hour period using excitation at 575 nm and measuring emission at 620 nm with a bandpass filter of 5 nm). The results are shown in **Response Figure 1**. Simply put, the MS2 protein does not have any auto-fluorescence at the wavelengths used to measure mRFP1 expression levels. As measured previously, the total amount of auto-fluorescence (called “background” fluorescence in our manuscript’s method section) is very small (10 to 20 fluorescence units), which has already been subtracted from our measurements to determine the mRFP1 expression levels.

It is not surprising that MS2 protein does not emit auto-fluorescence at 575 nm / 620 nm. Many proteins absorb light, particularly in the UV range. But the emission of fluorescence requires the presence of a specific structure within the protein (called a chromophore) that becomes excited by a photon and then re-emits a less energetic photon. This difference between absorption and fluorescence is a key reason why fluorescent protein reporters have become so useful in molecular & cellular biology.

The reviewer’s second question pertains to how we use the no-aptamer controls to measure the non-specific effects affecting protein expression levels. The reviewer suggests that we should consider the difference in fluorescence levels from the no-aptamer control measurements and subtract this difference from our riboswitch measurements. To answer this question, we carry out the analysis as the reviewer suggests for a MS2 riboswitch (ON switch), which will be instructive. In **Response Figure 2**, we first show the mRFP1 fluorescence levels of the riboswitch and the no-aptamer control as MS2 expression in increased, **without any normalization**.

Response Figure 2: The measured end-point mRFP1 fluorescence levels for the M2 riboswitch and the no-aptamer control when adding increasing amounts of MS2 expression plasmid.

Next, as the reviewer suggests, we carry out subtractive normalization, using the equation suggested by the reviewer (**Response Figure 3**). In the top panel, we show: $Y(x) = \text{FLU}_{\text{riboswitch}}(x) - [\text{FLU}_{\text{no-aptamer}}(x) - \text{FLU}_{\text{no-aptamer}}(x=0)]$, where $\text{FLU}_{\text{riboswitch}}(x)$ is the fluorescence level of the riboswitch measured at each MS2 expression plasmid concentration and $\text{FLU}_{\text{no-aptamer}}(x)$ is the fluorescence level of the no-aptamer control measured under the same conditions. In the bottom panel, we show the corresponding activation ratio, which is $Y(x) / Y(x=0)$.

Response Figure 3: The fluorescence levels and activation ratio of the M2 riboswitch if analyzed using the subtractive normalization approach suggested by the reviewer.

As one can see, even when using the reviewer’s suggested analysis approach, the M2 riboswitch reaches an activation ratio of 8.2-fold when 2 nM of MS2 expression plasmid is added, while the maximum activation ratio is 12.3-fold when adding 16 nM of MS2 expression plasmid. The M2 riboswitch strongly activates mRFP1 expression at low MS2 plasmid concentrations while the no-aptamer control’s mRFP1 expression levels barely change from 0 to 2 nM plasmid. However, the subtractive normalization approach does cause a large inconsistency in the data analysis when applied across the different types of riboswitches. Specifically, when the subtractive normalization is applied to the riboswitch OFF switches, some of the corrected mRFP1 expression levels become large negative numbers (not physically possible) and some repression ratios **become higher than we originally reported**.

We show that the correct way of analyzing the data can be derived from first principles. The reviewer is absolutely correct that the key objective of the normalization is to subtract the effects of the non-specific interactions. But the **key realization and distinction here** is that the **strengths of the interactions controlling translation rates** (and transcription rates) have a proportional effect **on the fluorescence measurements**. Specifically, the ribosome’s binding free energy to the mRNA (ΔG_{total}) controls the ribosome’s probability of binding to the mRNA and the mRFP1 expression level (r) according to Boltzmann’s relationship, which is:

$$r = K * \exp(-\beta * \Delta G_{\text{total}}) \quad \text{where } K \text{ is a proportionality factor and } \beta \text{ is a constant.}$$

This equation is central to statistical thermodynamics and we use it to predict the translation initiation rates of our mRNA sequences (the RBS Calculator model). We also use Boltzmann’s relationship to calculate transcription initiation rates (our Promoter Calculator model, LaFleur et. al. Nature Communications 2022). The proportionality factor lumps together several constant numbers (e.g. the plasmid DNA concentration). The key point here is that **a change in interaction strength will lead to a proportional change in the expression outcome**.

We can see that with a simple comparison. We measure mRFP1 expression levels of the no-aptamer control (r_1) with a corresponding ribosome-mRNA binding free energy of ΔG_1 (without any MS2 protein present). We then measure the mRFP1 expression levels of the no-aptamer control when adding some amount of MS2 protein (r_2) with a corresponding ribosome-mRNA binding free energy of ΔG_2 . We can use

these measurements to determine the strength of the non-specific interaction (ΔG_{NS}) via **subtraction of the binding free energies**:

$$\Delta G_{NS} = \Delta G_2 - \Delta G_1$$

We then apply Boltzmann's relationship and obtain that:

$$r_1 = K * \exp(-\beta * \Delta G_1)$$

$$r_2 = K * \exp(-\beta * \Delta G_2)$$

$$\Delta G_1 = -\log(r_1 / K) / \beta$$

$$\Delta G_2 = -\log(r_2 / K) / \beta$$

$$\Delta G_{NS} = \Delta G_2 - \Delta G_1 = -\log(r_2 / r_1) / \beta$$

$$(r_2 / r_1) = \exp(-\beta \Delta G_{NS})$$

These equations tell us that we can measure ΔG_{NS} (at a particular MS2 protein concentration) by using the ratio of the mRFP1 expression levels (r_2 / r_1) for the no-aptamer control. When MS2 protein is added and the no-aptamer control's translation rate increases ($r_2 > r_1$), then ΔG_{NS} has a negative value (ie, heat is released when the non-specific interaction takes place).

We now consider the same comparison for a riboswitch. We measure the mRFP1 expression levels of a riboswitch (r_3) with a corresponding ribosome-mRNA binding free energy of ΔG_3 (without any MS2 protein present). We then measure the mRFP1 expression levels of a riboswitch when adding the same amount of MS2 protein (r_4) with an apparent ribosome-mRNA binding free energy of ΔG_4 . **But a portion of this ΔG_4 is due to the presence of the non-specific interaction with strength ΔG_{NS} .** To remove this effect, we **subtract the non-specific interaction strength ($\Delta G_4 - \Delta G_{NS}$)**. Now the mRFP1 expression level of the riboswitch without the non-specific interaction is:

$$r_{4,corrected} = \exp(-\beta [\Delta G_4 - \Delta G_{NS}]) = \exp(-\beta \Delta G_4) / \exp(-\beta \Delta G_{NS}) = r_4 / (r_2 / r_1)$$

The activation ratio of the riboswitch without the non-specific interaction strength is:

$$AR = \exp(-\beta [\Delta G_4 - \Delta G_3 - \Delta G_{NS}]) = (r_4 / r_3) / (r_2 / r_1)$$

These equations tell us that we divide the riboswitch activation ratio by the no-aptamer control's activation ratio (r_2/r_1) – to remove the interaction strength of the non-specific interactions.

In **Figure 3**, we show model-predicted mRNA structural re-arrangements and free energy changes when the MS2 protein binds to a designed riboswitch, causing translation activation. In the example ON switch, ΔG_3 is -3.7 kcal/mol and ΔG_4 is -8.1 kcal/mol. The ribosome's binding free energy to the no-aptamer control is calculated to be $\Delta G_1 = -7.5$ kcal/mol.

Next, we consider the possibility that the addition of the MS2 protein somehow affects both translation rates and the mRNA level. We show that the same analysis approach should be used to remove this non-specific effect from the riboswitch's measured activation ratios. Again, the key principle here is that a change in mRNA concentration proportionally affects the measured mRFP1 expression levels.

We begin by rewriting our formula for the translation rate (r) while explicitly pulling the mRNA concentration out of the proportionality factor K:

$$r = K * [\text{mRNA}] * \exp(-\beta * \Delta G_{\text{total}}) \quad \text{where } [\text{mRNA}] \text{ is the mRNA concentration.}$$

We measure mRFP1 expression levels of the no-aptamer control (r_1) with a baseline mRNA concentration of $[\text{mRNA}]_1$. We then measure the mRFP1 expression levels of the no-aptamer control when adding some amount of MS2 protein (r_2) with a corresponding change in mRNA concentration, $[\text{mRNA}]_2$. The change in mRNA concentration, due to adding the MS2 protein, is measured by taking the ratio of r_2 over r_1 .

$$r_1 = K * [\text{mRNA}]_1 * \exp(-\beta * \Delta G_1) \quad r_2 = K * [\text{mRNA}]_2 * \exp(-\beta * \Delta G_2)$$

$$(r_2 / r_1) = ([\text{mRNA}]_2 / [\text{mRNA}]_1) * \exp(-\beta \Delta G_{\text{NS}}) \quad \Delta G_{\text{NS}} = \Delta G_2 - \Delta G_1$$

We now consider the same comparison for a riboswitch. We measure the mRFP1 expression levels of a riboswitch (r_3) with a baseline mRNA concentration of $[\text{mRNA}]_3$. We then measure the mRFP1 expression levels of a riboswitch when adding the same amount of MS2 protein (r_4) with a change in mRNA concentration, $[\text{mRNA}]_4$. **We would like to remove the change in mRNA concentration (due to the addition of MS2 protein) from the riboswitch's apparent activation ratio.** To do that, we divide the riboswitch's measured AR by the ratio $([\text{mRNA}]_2 / [\text{mRNA}]_1)$, which quantifies this non-specific effect. At the same time, we subtract ΔG_{NS} from the ribosome's binding free energy. Together, this yields:

$$r_{4,\text{corrected}} = ([\text{mRNA}]_2 / [\text{mRNA}]_1) * \exp(-\beta [\Delta G_4 - \Delta G_{\text{NS}}]) = r_4 / (r_2 / r_1)$$

The activation ratio of the riboswitch without the non-specific effects on $[\text{mRNA}]$ and ΔG_{NS} is:

$$\text{AR} = ([\text{mRNA}]_2 / [\text{mRNA}]_1) * \exp(-\beta [\Delta G_4 - \Delta G_3 - \Delta G_{\text{NS}}]) = (r_4 / r_3) / (r_2 / r_1)$$

Overall, we fully agree with the reviewer that non-specific interactions (any changes in translation rates or mRNA levels when the protein-of-interest is added) must be removed from the riboswitches' apparent activation or repression ratios. We show – from first principles – that the way to carry out this analysis is to divide the riboswitch's activation or repression ratios by the no-aptamer control's activation or repression ratios. Besides the first-principles derivation, we can see that this analysis approach is correct:

1. The corrected activation ratios have a sigmoidal relationship vs. the added protein concentrations. This is expected. We know that mRFP1 expression levels should increase sharply once the protein concentration exceeds the equilibrium dissociation constant of the RNA aptamer and then plateau to a maximum once all of the RNA aptamer is bound by protein. We see this behavior very clearly for the MS2-binding riboswitches where the RNA aptamer has high affinity to MS2 with a very small equilibrium dissociation constant (0.70 nM). This behavior is not seen when using the subtractive normalization approach (**Response Figure 3**).
2. For the riboswitch OFF switches, the corrected repression ratios are never negative. Adding more mCRP or IL-32 γ protein results in higher repression ratios. This is not true for the subtractive normalization approach, where the repression ratios for the mCRP OFF switches become negative at intermediate mCRP concentrations.
3. For both types of riboswitches, it did not matter if the addition of protein non-specifically affected either the translation rates or mRNA levels. In either or both scenarios, the analysis approach that we used will remove these non-specific effects. We do not need to differentiate or distinguish

between the sources of the non-specific interactions as they have identical effects on the mRNA expression levels.

4. This analysis approach is used by other research labs that specialize in studying or engineering riboswitches. Examples include: Tabuchi, T., & Yokobayashi, Y. (2022). High-throughput screening of cell-free riboswitches by fluorescence-activated droplet sorting. *Nucleic Acids Research*, 50(6), 3535-3550 and Van Vlack, E. R., Topp, S., & Seeliger, J. C. (2017). Characterization of engineered PreQ1 riboswitches for inducible gene regulation in mycobacteria. *Journal of Bacteriology*, 199(6), e00656-16.

Reviewer #2 (Remarks to the Author):

The authors have addressed the key concerns raised in the original round of review in a satisfactory manner.

We thank the reviewer for their supportive comments.

Reviewer #1 (Remarks to the Author):

For this reviewer, the major concerns have not sufficiently been addressed. The major concerns are still present. In detail:

- Fluorescent readouts in biological systems relying on translation and transcription are not solely determined by translation initiation rates. This is an inadequate simplification of complex biological systems.
- Consequently, riboswitch M2 can more truthfully be described by the dose response curve as presented in Response Figure 3.
- Sequence C5 as shown in Figure 4 of the manuscript is reported misleadingly and does not behave as one expects from a translational riboswitch. The "no aptamer" control values (supplementary data set, sheet "Riboswitch Dosage Curves") shows enormous unspecific effects that cannot be corrected for as proposed by the authors. Therefore, there is no regulation at all.
- For sequence I2 there is no "no aptamer" control provided for the dose response and as such not valid. Since dramatic unspecific effects are reported for M2 and C5, the same must be expected for I2.
- Finally, regarding the closing remark "4." in the rebuttal, neither of the cited papers appear to use the analysis approach used by the authors.

Reviewer #3 (Remarks to the Author):

The manuscript has been the subject of concerns related to the reporting of riboswitch function. The authors have employed a data presentation method that involves multiple rounds of normalization to obtain an activation/repression ratio. However, a reviewer has expressed doubt about the accuracy of this analysis and believes it could lead to misinterpretation of the results.

Having reviewed the letter, I find that the determination of the activation/repression ratio is generally acceptable, but there are still some issues to be addressed. Firstly, this analysis poses challenges for some of the data. Secondly, I believe that the authors should present the unnormalized version of the data in the main figure to allow readers to draw their own conclusions.

In relation to Figure 4, I have included a modified version of the graph that shows the unnormalized data and the figure provided by the authors. The left and right panels appear to be reasonable, although it would be nice to have a no aptamer control for the right panel and to use the same normalization method for data. The middle panel is more concerning as the repression ratios are concealing important experimental details that could influence the interpretation of the data. It is evident from the raw data that hCRP is highly toxic to the reaction, which is not reflected in the paper's figure. Furthermore, the presentation in the paper fails to convey that the condition of greatest repression (i.e., 2500) is under conditions when the reaction is essentially dead. As this condition and its corresponding repression ratio is called out in the abstract, I believe the full presentation of the data is appropriate.

Figure 4. Dose-response of MS2, CRP, and IL32 γ riboswitches. The activation or repression ratios of the (A) M2, (B) C5, and (C) I2 riboswitch sensors in response to varied concentrations of phage MS coat protein, human mCRP protein, and human IL32 γ protein, respectively. Dots and bars are the mean and standard deviation of replicate cell-free assays (M2, N = 6 biological replicates; C5, N = 6 biological replicates; I2, N = 8 biological replicates).

Title: Automated Design of Protein-binding Riboswitches for Sensing Human Biomarkers in a Cell-free Expression System

Authors: Grace E. Vezeau, Lipika R. Gadila, Howard M. Salis

We would like to express our gratitude to reviewer #3 for taking the time to read our manuscript and for providing objective and specific recommendations to resolve any ambiguities or concerns. Below, we described the changes made to the manuscript that resolve such ambiguities. We also directly address reviewer #2's remaining concerns. Our responses are written in **blue** text.

Reviewer #3:

The manuscript has been the subject of concerns related to the reporting of riboswitch function. The authors have employed a data presentation method that involves multiple rounds of normalization to obtain an activation/repression ratio. However, a reviewer has expressed doubt about the accuracy of this analysis and believes it could lead to misinterpretation of the results. Having reviewed the letter, I find that the determination of the activation/repression ratio is generally acceptable, but there are still some issues to be addressed. Firstly, this analysis poses challenges for some of the data. Secondly, I believe that the authors should present the unnormalized version of the data in the main figure to allow readers to draw their own conclusions. In relation to Figure 4, I have included a modified version of the graph that shows the unnormalized data and the figure provided by the authors. The left and right panels appear to be reasonable, although it would be nice to have a no aptamer control for the right panel and to use the same normalization method for data. The middle panel is more concerning as the repression ratios are concealing important experimental details that could influence the interpretation of the data. It is evident from the raw data that hCRP is highly toxic to the reaction, which is not reflected in the paper's figure. Furthermore, the presentation in the paper fails to convey that the condition of greatest repression (i.e., 2500) is under conditions when the reaction is essentially dead. As this condition and its corresponding repression ratio is called out in the abstract, I believe the full presentation of the data is appropriate.

We agree with the reviewer and have updated **Figure 4** to show the measured mRFP1 fluorescence levels for the riboswitches alongside the no-aptamer control measurements, which clearly show that the riboswitches are regulating mRFP1 expression in response to increasing concentrations of the protein ligand. The no-aptamer control is a construct that constitutively expressed mRFP1 at a moderately high expression level whereas the riboswitches were designed to either activate or repress mRFP1 expression (at the translation step) in response to protein ligand binding. Next to this plot, we show the normalized mRFP1 fluorescence levels (X / X_0 , where X_0 is the mRFP1 fluorescence level at zero protein concentration) to clearly show the change in mRFP1 expression levels for riboswitches and the no-aptamer control as protein concentrations vary. Next to this plot, we show the activation or repression (fold-change) for each riboswitch, where we use the no-aptamer control data to remove the non-specific effects. Readers can clearly see that the riboswitches and no-aptamer controls have distinctly different dosage responses, supporting our conclusions. All data are available in numerical form in our original Supplementary Data.

We have updated the text in the manuscript to discuss these new plots and to highlight a few key points.

1. We point out that a 2500 nM mCRP concentration inhibited cell-free expression, increasing the variability of the riboswitch sensor measurement as quantified in the standard deviation.
2. When using these riboswitch sensors in a future diagnostic device, the no-aptamer control measurements would always be performed on the same sample at the same time. The reported

measurement of the protein concentration will depend on both measurement outcomes. The no-aptamer control data provides useful information to ensure that the assay was performed correctly and non-specific effects on cell-free gene expression are well-characterized and quantified. We use this data to quantify how well the riboswitch activated or repressed translation rate in order to validate and compare riboswitch designs. As different proteins will exhibit different non-specific effects on cell-free expression systems, this approach provides a unified way of answering two key questions: “By how much did the added protein affect cell-free expression?” and “By how much did the designed riboswitch activate or repress translation rate?”.

3. The dosage response for the no-aptamer controls is often non-linear (e.g. low mCRP concentrations increase cell-free expression whereas high mCRP concentrations inhibit cell-free expression). In contrast, the designed riboswitches activate or repress translation rates according to linear or sigmoidal responses, which is expected according to the known biophysics of riboswitch regulation. The dosage responses for no-aptamer controls versus riboswitches can proceed in the same direction (e.g. MS2 and the M2 ON switch) or in opposite directions (mCRP and the C5 OFF switch), depending on the protein ligand and the designed riboswitch. There is no right or wrong scenario here; the main criterion for a good sensor is that we can sufficiently distinguish the sensor outcome from the baseline control outcome at each relevant protein concentration (ie, maximum information content). Modern diagnostic devices can record, quantify, and digitize multiple output signals, enabling precise analysis and rigorous statistics, carried out on the device.

Reviewer #2:

Fluorescent readouts in biological systems relying on translation and transcription are not solely determined by translation initiation rates. This is an inadequate simplification of complex biological systems. Consequently, riboswitch M2 can more truthfully be described by the dose response curve as presented in Response Figure 3.

We agree that the both transcription rates and translation rates collectively control protein expression levels. As described in the manuscript, our model of riboswitch regulation enables us to design synthetic riboswitches that activate or repress translation initiation rates. A key purpose of our no-aptamer controls is to ensure that our mRFP1 protein expression level measurements can be used to quantify the change in translation rate. The “non-specific effects” as mentioned in the manuscript include non-specific changes in transcription rate. By removing these non-specific effects during the data analysis, we focus on answering the question “By how much did the riboswitch activate or repress translation initiation rate?”.

Sequence C5 as shown in Figure 4 of the manuscript is reported misleadingly and does not behave as one expects from a translational riboswitch. The “no aptamer” control values (supplementary data set, sheet “Riboswitch Dosage Curves”) shows enormous unspecific effects that cannot be corrected for as proposed by the authors. Therefore, there is no regulation at all.

Riboswitch C5 represses translation initiation rate with a linear-like response to increasing mCRP concentrations. It behaves exactly as expected according to the biophysics of riboswitch-regulated translation rates as quantified and predicted by our model of riboswitch regulation. The no-aptamer control is measuring the non-specific changes in cell-free expression as mCRP concentration varies, which can include multiple competing effects (some positive and some negative). As such, the no-aptamer control response can be non-linear as we see.

For sequence I2 there is no “no aptamer” control provided for the dose response and as such not valid. Since dramatic unspecific effects are reported for M2 and C5, the same must be expected for I2.

As stated in the original manuscript, we carried out no-aptamer control experiments while changing IL32 γ concentrations. We did not observe a statistically significant change in no-aptamer control expression levels between 0 and 780 nM (the extremes of the tested range). We have highlighted this fact in the updated manuscript text. This data is now included in the updated Figure 4.